

# Definition differences and internal variability affect the simulated Arctic sea ice melt season

Abigail Ahlert[1] and Alexandra Jahn[1]

[1]Department of Atmospheric and Oceanic Sciences and Institute of Arctic and Alpine Research, University of Colorado Boulder, Boulder, USA

**Correspondence:** Abigail Ahlert (abigail.ahlert@colorado.edu)

**Abstract.**

Satellite observations show that the Arctic sea ice melt season is getting longer. This lengthening has important implications for the Arctic Ocean's radiation budget, marine ecology and accessibility. Here we assess how passive microwave satellite observations of the melt season can be used for climate model evaluation. By using the Community Earth System Model Large Ensemble (CESM LE), we evaluate the effect of multiple possible definitions of melt onset, freeze onset and melt season length on comparisons with passive microwave satellite data, while taking into account the impacts of internal variability. We find that within the CESM LE, melt onset shows a higher sensitivity to definition choices than freeze onset, while freeze onset is more greatly impacted by internal variability. The CESM LE accurately simulates that the trend in freeze onset largely drives the observed pan-Arctic trend in melt season length. Under RCP8.5 forcing, the CESM LE projects that freeze onset dates will continue to shift later, leading to a pan-Arctic average melt season length of 7–9 months by the end of the 21st century. However, none of the available model definitions produce trends in the pan-Arctic melt season length as large as seen in passive microwave observations. This suggests a model bias, which might be a factor in the generally underestimated response of sea ice loss to global warming in the CESM LE. Overall, our results show that the choice of model melt season definition is highly dependent on the question posed, and none of the definitions exactly matches the physics underlying the passive microwave observations.

## 1 Introduction

Arctic sea ice melt season characteristics play an important role in the radiation balance of the Arctic. Changes in the melt season have important implications for the Arctic climate system as a whole (Markus et al., 2009), and therefore are crucial for anticipating ecological changes and informing economic development in the region. In this study, we quantify the impact of definition choices and internal variability on Arctic sea ice melt season characteristics (averages and trends of melt onset, freeze onset and melt season length). This allows us to assess how best to compare observed and modeled melt season changes and diagnose model biases.





As ice retreats during the summer months, incident solar radiation is absorbed by the Arctic Ocean. The timing of ice retreat and advance, here referred to as melt onset and freeze onset, therefore greatly affects regional oceanic heat budgets (Bitz and Roe, 1996; Perovich et al., 2007, 2011; Stroeve et al., 2014). In situ observations indicate that melt onset is driven primarily by synoptic frontal systems that produce northward warm air advection (Persson, 2012). Cloud formation and light drizzle in

the warm air layer then increase downwelling longwave radiation and initiate melt onset. For each day that melt onset occurs earlier, 8.7 MJ/m$^2$ is absorbed by the surface (Perovich et al., 2007). As summer energy absorption increases with earlier melt onset dates, positive ocean heat content anomalies in the near-surface temperature maximum (NSTM) layer increase in magnitude (Timmermans, 2015). At the end of the summer, the heat stored in the NSTM layer is then mixed toward the surface by shear-driven mixing and entrainment, delaying freeze onset (Perovich et al., 2007; Stroeve et al., 2012; Steele et al., 2008).

For each day that freeze onset occurs later, an additional 1.5 MJ/m$^2$ is absorbed by the surface (Perovich et al., 2007).

Relationships between sea ice extent and melt season length (Stroeve et al., 2014), and specifically between sea ice extent and melt onset date (Wang et al., 2011), have been found. Furthermore, because the timing of melt onset has a large impact on radiation absorption in the Arctic, observed melt onset dates have been used to predict freeze onset dates in some regions, such as Baffin Bay and the Laptev and East Siberian seas (Stroeve et al., 2016). The existence of these relationships raises the

possibility that melt season biases might be contributing to biases in sea ice extent simulations.

Melt and freeze onset dates also have important ecological and societal implications in the Arctic. For example, delayed freeze onset has been shown to decrease snowpack on sea ice, thereby reducing the area that ringed seals can use for snow caves necessary for birthing (Hezel et al., 2012). Polar bears are also dependent on the timing of melt and freeze onset, as they use sea ice as a platform for seasonal hunting and breeding (Stern and Laidre, 2016). Moreover, prediction of melt onset dates

is increasingly important for operational sea ice forecasts that inform local decision-making in the Arctic (Collow et al., 2015).

Previous efforts to assess melt onset, freeze onset and melt season length in GCMs have used a variety of definitions, as no model definition of melt and freeze onset directly corresponds to remote sensing definitions (Wang et al., 2011; Jahn et al., 2012; Mortin and Graversen, 2014; Holland and Landrum, 2015; Johnson and Eicken, 2016; Wang et al., 2017). This inconsistent definition of melt and freeze onset complicates both comparisons between models and between models and observations.

Furthermore, because of the chaotic nature of the climate system, there will always be a limit to how well model projections fit observations, even for 35+ year trends (Kay et al., 2015; Notz, 2015; Swart et al., 2015). In particular, it has been shown that the full CMIP5 distribution of 35-year September sea ice extent trends could be due to internal variability (Swart et al., 2015). Recent work suggests that sea ice trends similar to observations are only found in GCMs with too much global warming (Rosenblum and Eisenman, 2017). But global warming, known to drive sea ice extent trends (Mahlstein and Knutti, 2012), is

also strongly impacted by internal variability (Jahn, 2018). Furthermore, observational estimates of the sea ice sensitivity are highly uncertain due to observational uncertainties in both sea ice extent and global temperature (Niederdrenk and Notz, 2018).

Further complicating the issue, even observational assessments of melt season characteristics do not use just one definition of melt onset and freeze onset. Passive remote sensing techniques utilize brightness temperatures, which are sensitive to the changes in emissivity that occur when snow and ice change phase. The Markus et al. (2009) passive microwave algorithm

(PMW) determined pan-Arctic early melt and freeze onset dates as well as continuous melt and freeze onset dates over the



period 1979–2008. Stroeve et al. (2014) extended the Markus et al. (2009) dataset to 2013. Other algorithms exist in addition to the PMW algorithm, such as the advanced horizontal range algorithm (AHRA). AHRA computes melt onset (but not freeze onset) over both first year ice and multiyear ice based on passive microwave temperatures (Drobot and Anderson, 2001), building upon earlier work that only provided melt and freeze onset dates over multiyear sea ice (Smith, 1998). Surface air

temperatures from the IABP/POLES dataset were then incorporated into the AHRA to better constrain melt onset dates and produce freeze onset dates (Belchanksy et al., 2004). While both the PMW and AHRA algorithms utilize passive microwave brightness temperatures, they are not equally sensitive to changes in brightness temperatures. Comparison between the AHRA and the PMW algorithms show large mean differences in melt onset dates and statistically significant differences in trends over 1979–2012 (Bliss et al., 2017). When reproduced with the same inter-sensor calibration adjustments and masking techniques,

trend agreement improves between PMW and AHRA, but large differences in mean melt onset dates remain (Bliss et al., 2017).

All of this raises two main questions: What are the impacts of different definition choices and internal variability on diagnosing and projecting Arctic sea ice melt season characteristics (melt onset, freeze onset and melt season length)? How can we use melt season characteristics from satellite observations for model evaluation, despite those effects? We seek to answer these questions by using the longest available satellite-derived melt and freeze onset data set (Stroeve et al., 2014) to compare

multiple plausible definitions of melt and freeze onset in the Community Earth System Model Large Ensemble (CESM LE) (Kay et al., 2015). By using the CESM LE, we are able to account for the role of internal variability and utilize daily model variables that are not available from the CMIP5 archive, thereby allowing us to assess the comparability of different melt and freeze onset definitions. We also show how melt and freeze onset dates and melt season length are projected to change by the end of the 21st century under a strong emission scenario (RCP8.5) and how internal variability and definition differences

impact those projections.

## 2   Methods

In this study, we use both model and PMW satellite data to assess the timing of continuous sea ice melt and freeze onset in the Arctic, defined here as north of 66°N.

### 2.1   Community Earth System Model Large Ensemble

To analyze the impact of different model definitions and internal variability on melt season characteristics, we use the CESM LE (Kay et al., 2015). The CESM LE is a 40-member ensemble of simulations conducted for the period 1920–2100. Each ensemble member starts from slightly different initial atmospheric conditions and is subject to historical forcing from 1920–2005 and RCP8.5 forcing from 2006–2100. The CESM LE uses CESM1-CAM5, and has a nominal resolution of 1°× 1°. The CESM LE has been used in multiple studies of Arctic sea ice cover, performing well overall (Swart et al., 2015; Barnhart

et al., 2016; Jahn et al., 2016; Rosenblum and Eisenman, 2017; Jahn, 2018; Massonnet et al., 2018; Labe et al., 2018). Under RCP8.5 forcing, Arctic sea ice in the CESM LE first reaches September ice-free conditions by the middle of the 21st century





(2032–2053 using monthly means of ice extent, Jahn et al., 2016). By the end of the 21st century, ice-free conditions persist for 4–5 months in most years (Jahn, 2018).

## 2.2 Passive microwave melt and freeze onset data

To assess the CESM LE melt season characteristics, we utilize the PMW dataset of melt and freeze onset dates from Markus et al. (2009), updated by Stroeve et al. (2014), gridded to 25 km x 25 km. This dataset applies the PMW melt and freeze onset algorithm to passive microwave brightness temperatures collected from the Nimbus 7 scanning multichannel microwave radiometer (SSMR), the Special Sensor Microwave/Imager (SSM/I), and the Special Sensor Microwave Imager and Sounder (SSMIS). Brightness temperatures from the 37V GHz and 19V GHz (18V GHz on SSMR) sensor channels are used to produce both early melt/freeze onset dates and continuous melt/freeze onset dates from 1979–2014. The former is defined as the first day of melt/freeze, while the latter is the day that melting or freezing conditions begin and persist throughout the rest of the season.

Because passive microwave brightness temperatures can be used to derive melt and freeze onset dates across the entire Arctic for a 36-year period, these data are well suited for GCM evaluation. In this study, we use the continuous melt and freeze onset dates so that we can determine the continuous melt season length. All further discussion of melt and freeze onset refers to continuous melt and freeze onset.

## 2.3 Model definitions of melt and freeze onset

Because GCMs, including the CESM LE, do not simulate brightness temperatures, we cannot apply the same methodology as used in the PMW algorithm to define melt and freeze onset in the model. Instead, we define several melt and freeze onset dates from the existing daily output of the CESM LE that make physical sense to assess the importance of definition choices and their suitability for comparisons with the PMW data. Details of the definitions can be found in Table 1. In particular, we make use of the daily fields of snowmelt, surface temperature, frazil and congelation ice growth, and thermodynamic ice volume tendency. Of these, only surface temperature and thermodynamic ice volume tendency are available for all 40 ensemble members. All others were only saved for two ensemble members (34 and 35).

For melt onset in the CESM LE, we create three different definitions, based on the available output (Table 1): one definition using thermodynamic ice volume tendency (for all 40 members), a second using surface temperature where ice concentration is greater than zero (for all 40 members), and a third definition using snowmelt (for two members). We expect that the snowmelt definition matches the PWM definition most closely, as the brightness temperature melt criteria captures changes in liquid water content in the snow. The temperature criteria likely also captures snowmelt onset, but less directly than if melt onset is based on actual snowmelt. In contrast, the thermodynamic volume tendency captures the onset of ice-melt rather than snow melt. However, as the PMW is based on liquid water content in the snowpack, and the snowmelt definition is due to snowmelt itself, even the snowmelt melt onset definition likely does not correspond perfectly to the PMW-based definition. Furthermore, snowmelt is only saved in two ensemble members, which does not allow an assessment of the impact of internal variability



| Definition Names | Output Variable in the CESM | Threshold | Consecutive Days |
|---|---|---|---|
| **Melt Onset** | | | |
| Surface Temperature | TS | -1 °C | 3 |
| Thermodynamic Ice Volume Tendency | dvidtt_d | 0 cm/day | 3 |
| Snowmelt | melts_d | 0.01 cm/day | 5 |
| **Freeze Onset** | | | |
| Surface Temperature | TS | -1.8 °C | 21 |
| Thermodynamic Ice Volume Tendency | dvidtt_d | 0 cm/day | 3 |
| Congelation Ice Growth | congel_d | 0.01 cm/day | 3 |
| Frazil Ice Growth | frazil_d | 0 cm/day | 3 |

**Table 1.** CESM LE definitions for melt and freeze onset, showing the model output variable name used, the threshold used and the number of consecutive days over which the variable must exceed the threshold for each definition. Details on how these thresholds and consecutive days were chosen can be found in the Supplementary.

on this definition. We will compare all three definitions in order to quantify how the diagnosed melt onset in the model varies based on the variable used.

For freeze onset in the CESM LE, we create four different definitions (Table 1): One using thermodynamic ice volume tendency (for all 40 members) and a second using surface temperature where ice concentration is greater than zero (for all 40 members). In the CESM LE, thermodynamic ice volume tendency is a sum of congelation ice growth along existing sea ice and frazil ice growth in the water colum. We therefore create two additional freeze onset definitions using frazil ice growth and congelation ice growth, in order to compare the impact of these two processes.

Melt season length is calculated at each grid cell for each year as the difference between local freeze onset date and melt onset date. In total, we create five unique definitions of melt season length, which are detailed in Table 2. Two definitions of melt season length keep like variables together (i.e. use both melt and freeze onset dates from surface temperature definitions and thermodynamic volume tendency) while the other three combine variable definitions (e.g., use melt onset dates from the

| Melt season length definition name | Melt onset definition | Freeze onset definition |
|---|---|---|
| Volume – Volume | Thermodynamic Ice Volume Tendency | Thermodynamic Ice Volume Tendency |
| Temperature – Temperature | Surface Temperature | Surface Temperature |
| Congelation – Snowmelt | Snowmelt | Congelation ice growth |
| Frazil – Snowmelt | Snowmelt | Frazil ice growth |
| Temperature – Snowmelt | Snowmelt | Surface temperature |

**Table 2.** CESM LE definitions for melt season length, showing the various melt and freeze onset definition combinations used to compute melt season length. For each combination, the melt onset date is subtracted from the freeze onset date at each grid cell every year.

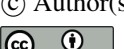



snowmelt definition and freeze onset dates from the frazil ice growth definition) in order to span the full range of possible melt season length definitions in the CESM.

Three key definition decisions were found to impact the melt and freeze onset definitions in the CESM LE: 1) The period over which one should check for melt and freeze onset, 2) the threshold each variable must meet for melt and freeze onset, and 3) the number of consecutive days each definition must pass the threshold for melt and freeze onset to occur. The choices are shown in Table 1. Decisions on these three components were based on what makes physical sense, whether they provide sensible continuous melt and freeze onset dates and the percent area of the Arctic where melt and freeze onset conditions are met. Details on the reasons for each of these choices can be found in the Supplementary. We did not use any smoothing techniques such as running means or medians, which were used in other studies (Mortin and Graversen, 2014; Holland and Landrum, 2015). We found that smoothing techniques excessively reduce the number of times that the melt and freeze onset criteria are met in the CESM LE, at least for some variables. Details can be found in the Supplementary.

## 3 Results

### 3.1 CESM LE definitions: average melt season characteristics

#### 3.1.1 Pan-Arctic averages

Using the definitions described in Sect. 2.3, we find that there are large differences in the pan-Arctic averages of melt season characteristics between CESM LE definitions (Fig. 1). To quantify pan-Arctic definition differences, we define the spread as the average difference between the earliest and latest melt and freeze onset definitions over 1979–2014, as well as the difference between the shortest and longest melt season length definitions over this time period. Here we discuss only ensemble member 35, as differences in spreads between ensemble members 34 and 35 are small (Fig. 1 and S.1). We find that the spread in pan-Arctic melt onset definitions in the chosen ensemble member is 35 days, due largely to the early melt onset dates from the thermodynamic ice volume tendency definition, which captures ice melt (including bottom melt). This spread in melt definitions is much larger than the 13 day spread found between the freeze definitions. The large spread in melt onset dates also affects differences between melt season length definitions, leading to a spread of 43 days in ensemble member 35. Note that spreads in pan-Arctic melt and freeze onset do not sum to the spread in melt season length, as the melt season length is calculated at each grid cell and not as a difference in the pan-Arctic means.

Internal variability introduces additional differences in diagnosed pan-Arctic melt onset, freeze onset, and melt season length (Fig. 2). However, these are much smaller than the definition spreads, ranging between 4-8 days. Average melt onset dates are less impacted by internal variability than average freeze onset dates, based on the temperature and thermodynamic ice volume tendency definitions where all 40 ensemble members are available. Pan-Arctic melt onset dates fall within a range of 5 days, while pan-Arctic freeze onset dates fall within a range of 8 days. Average melt season length is affected by internal variability similarly to average freeze onset dates, with a range of 7 days in both definitions.

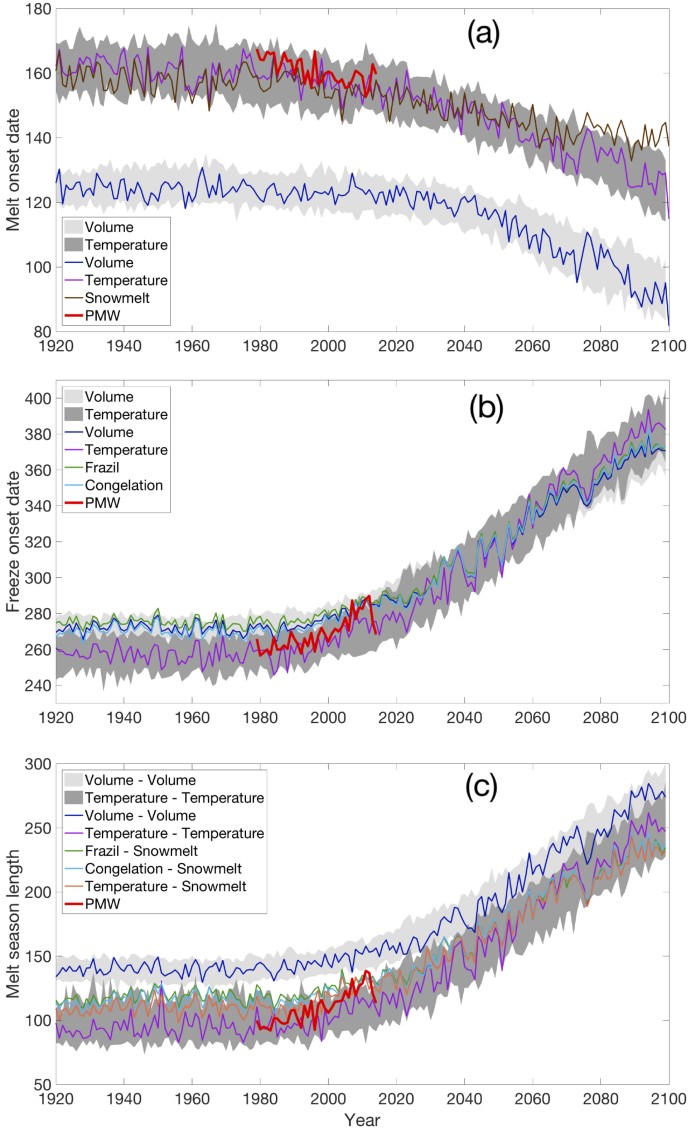

**Figure 1.** Melt season characteristics averaged over 66°N to 84.5°N for PMW satellite observations and each CESM LE definition for (a) melt onset (b) freeze onset and (c) melt season length. PMW satellite observations are shown in red. Other colored lines represent ensemble member 35, and the gray shading represents the ensemble spread for the two definitions (surface temperature and thermodynamic ice volume tendency) that have 40 ensemble members. Plots are reproduced with member 34 in colored lines in Fig. S.1.

### 3.1.2 Spatial averages

Areas in the marginal ice zone have earlier melt onset dates and later freeze onset dates than those in the Central Arctic, but specific spatial distributions of average melt season characteristics in the CESM LE depend on the definition. For example,



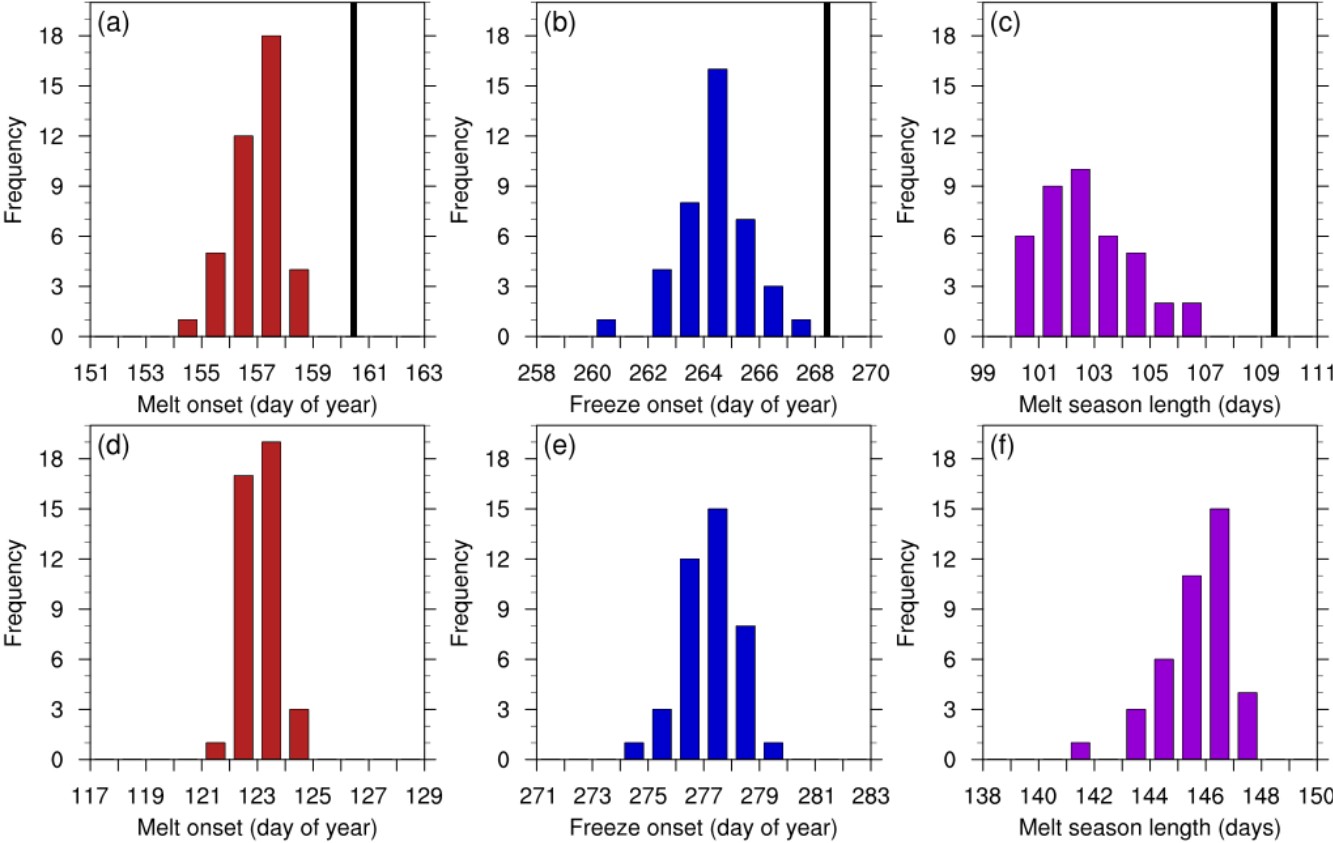

**Figure 2.** Histograms of the pan-Arctic average melt season characteristics over 1979–2014 using the surface temperature definitions (a–c) and thermodynamic ice volume tendency definitions (d–f) for all 40 CESM LE ensemble members, showing the impact of internal variability. PMW observations are denoted by black lines (a–c). Note that the x-axis limits are different in each panel, but the range is the same (12 days), to facilitate the assessment of the impact of internal variability for different processes and definitions.

melt onset derived from the snowmelt definition occurs in mid-to-late June in the Central Arctic and parts of the Laptev Sea (Fig. 3a). Melt onset dates in the surface temperature definition are generally later than in the snowmelt definition (Fig. 3b), with mid-to-late June melt onsets stretching from the Central Arctic into the East Siberian, Chukchi and Beaufort Seas. The thermodynamic ice volume tendency melt onset definition yields Central Arctic melt onset dates about 10 days earlier than the other definitions, as well as earlier onset dates in the Barents and Chukchi Seas (Fig. 3c). These differences are partially due to basal ice melt driven by the inflow of warm Atlantic/Pacific water, which is a melt process not captured by surface temperature or snowmelt-derived definitions of melt onset. Averages of melt season characteristics over 1979–2014 are similar for ensemble members 34 (shown in Fig. S.2–S.4) and 35 (Fig. 3–5), as the impact of internal variability on the 36-year means of the selected variables is small.





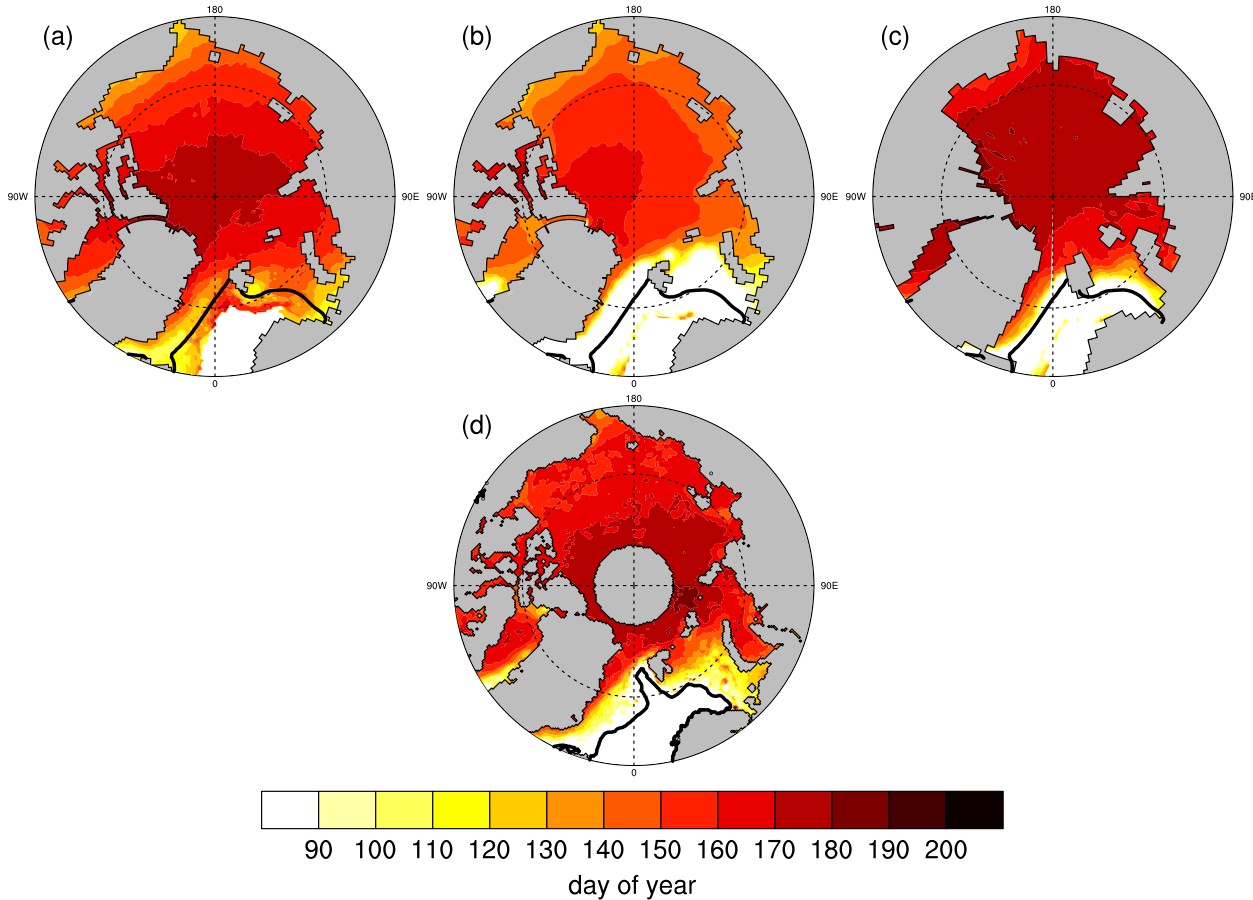

90 100 110 120 130 140 150 160 170 180 190 200
day of year

**Figure 3.** Average melt onset dates over 1979–2014 for each CESM LE definition using ensemble member 35: (a) snowmelt definition (b) thermodynamic ice volume tendency definition (c) surface temperature definition and (d) PMW satellite observations. The black line denotes the mean March ice edge (15% ice concentration) from 1979–2014 using (a–c) the CESM LE and (d) NSIDC Bootstrap sea ice concentrations (Comiso, Dataset accessed 2018-05-12). Melt onset dates south of the mean ice edge are less reliable than those north of the edge. Plots from ensemble member 34 are very similar and shown in Fig. S.2.

Average freeze onset dates over the satellite era also vary spatially by definition (Fig. 4a-e). In the Central Arctic, the surface temperature definition yields freeze onset dates in early-to-mid August. Freeze onset definitions based on sea ice variables also show early-to-mid August freeze onset dates in the region north of the Canadian Arctic and Greenland, but later freeze onset dates throughout the rest of the Central Arctic. In all definitions, there are large gradients in freeze onset in the marginal seas.

5 For example, in the Chukchi Sea, which is impacted by Pacific water inflow, freeze onset occurs between mid-September and the end of November. Even larger gradients exist in the Barents Sea, which is impacted by Atlantic inflow. Large gradients in



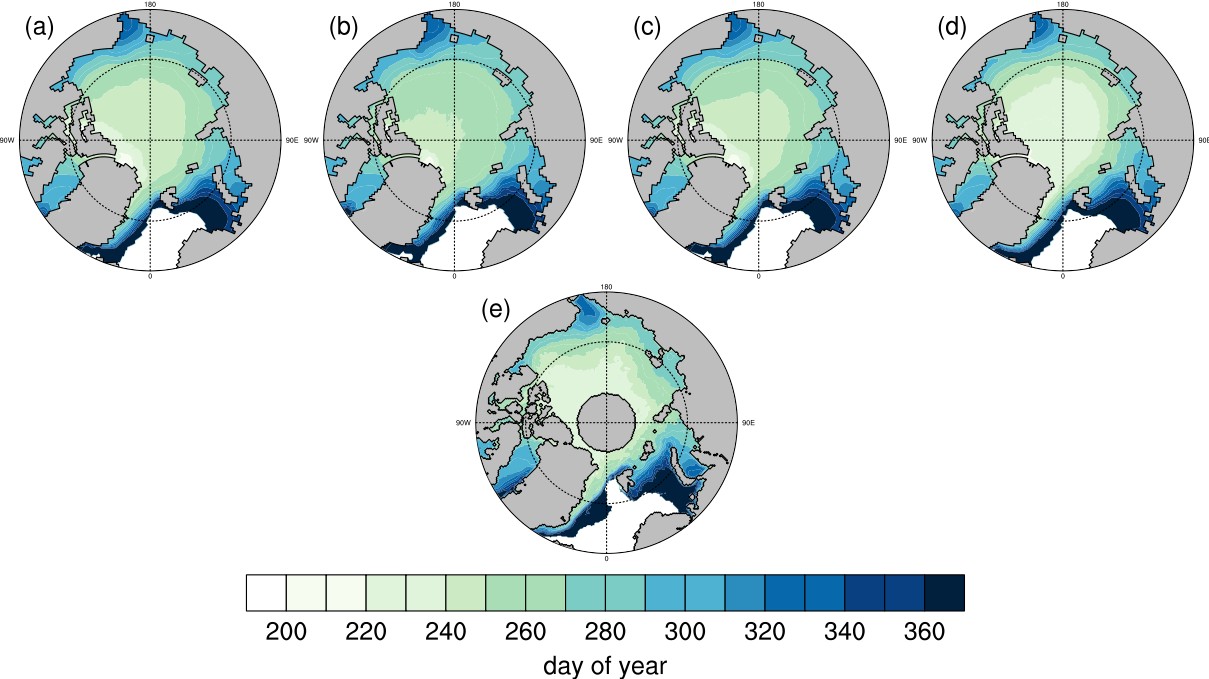

**Figure 4.** Average freeze onset dates for 1979–2014 for each CESM LE definition using ensemble member 35: (a) congelation ice growth definition (b) frazil ice growth definition (c) thermodynamic ice volume tendency definition (d) surface temperature definition and (e) PMW satellite observations. Plots from ensemble member 34 are very similar and shown in Fig. S.3.

the marginal ice zones are expected, as these areas show the largest trends in winter ice loss and are impacted most strongly by sensible and latent heat fluxes (Deser et al., 2000).

As expected, all definitions show the shortest melt seasons in the Central Arctic and the longest melt seasons in the marginal seas. Melt seasons along the Atlantic ice edge and in the Barents Sea are particularly long relative to the other marginal seas

5 (Fig. 5). However, the previously discussed differences in melt and freeze onset dates between definitions are noticeable when comparing definitions of melt season length. For example, thermodynamic ice volume tendency melt onset dates (which occur earlier than in the other definitions) drive the longer melt season lengths found along the Atlantic ice edge and in the Barents Sea when using the Volume - Volume definition (Fig. 5c). Additionally, in the Laptev Sea, surface temperature melt onset dates are later than those from the other definitions, and this drives shorter melt season lengths in the Temperature - Temperature

10 definition than the other CESM LE definitions by about 25 days (Fig. 5e).





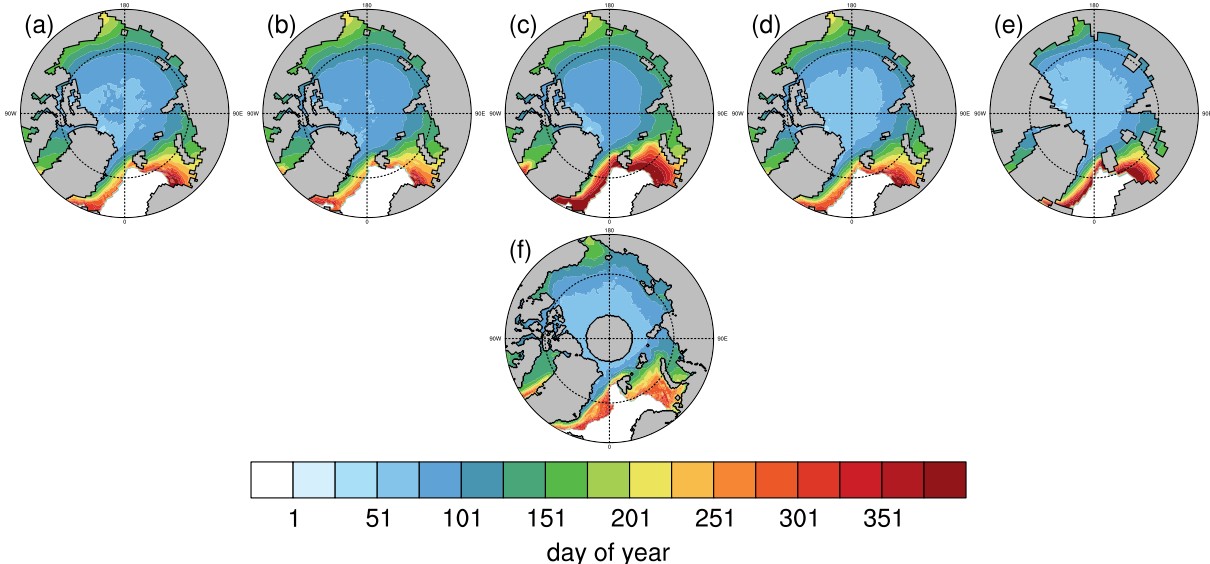

**Figure 5.** Average melt season lengths over 1979–2014, for each CESM LE definition using ensemble member 35: (a) Congelation - Snowmelt (b) Frazil - Snowmelt (c) Volume - Volume (d) Temperature - Snowmelt (e) Temperature - Temperature and (f) PMW satellite observations. Plots from ensemble member 34 are very similar and shown in Fig. S.4.

## 3.2 CESM LE definitions: trends in melt season characteristics

### 3.2.1 Pan-Arctic trends

Definitions in the CESM LE generally show pan-Arctic melt onset dates trending earlier and pan-Arctic freeze onset trending later over the period 1979–2014 (Table 3, Fig. 6), in agreement with previous work (Stroeve et al. (2014); Mortin and Graversen (2014); Johnson and Eicken (2016)). But in the CESM LE, internal variability affects the magnitude of these 36-year trends, and in a few cases for melt onset and melt season length even the sign of the trends. The large effect of internal variability on these trends is already evident when comparing trends between ensemble members 34 and 35 (Table 3). Ensemble member 35 shows larger pan-Arctic trends than ensemble member 34 over 1979–2014 for almost all model definitions and melt season characteristics. The only exception is the trend in melt onset derived from thermodynamic ice volume tendency, which is the smallest trend in both ensemble members, and shows a negative trend in member 34 but a small positive trend in member 35 (Table 3). The impact of internal variability on the 1979-2014 melt onset trends is even more pronounced using the full 40-member CESM ensemble, where melt onset trends fall between -2.4 and 0.8 days/decade for the surface temperature and thermodynamic volume tendency definitions (Fig. 6). However, all members show negative 36-year melt onset trends for the rest of the model simulation if we shift the trend start year to 1990 for the surface temperature definition and to 2008 for



the volume tendency definition. This shows that forced melt onset trends over the observed period can be masked by internal variability for some of the definitions of melt onset in the model.

Pan-Arctic freeze onset trends in the CESM LE are larger than trends in melt onset in all forty ensemble members, regardless of definition, and are always positive over the satellite era (indicating later freeze onset). Thirty-six year trends in freeze onset are positive throughout the remainder of the model simulation as well. The surface temperature definition of freeze onset yields the largest trend over the satellite era in ensemble members 34 and 35 (Table 3). The maximum trend of all ensemble members is also larger in the surface temperature definition than in the thermodynamic volume tendency definition (Table 3). In Figure 2, the pan-Arctic average freeze onset dates are more affected by internal variability than the averages melt onset dates. This is true for the pan-Arctic trends as well: there is greater variability between ensemble members in the freeze onset trends than in the melt onset trends (Fig. 6).

| | Member 34 trends | Member 35 trends | Ensemble minimum | Ensemble maximum | PMW Observations |
|---|---|---|---|---|---|
| **Melt Onset** | | | | | |
| PMW Observations | | | | | -2.5 |
| CESM LE Surface Temperature | -0.9 | -1.9 | -2.4 | 0.8 | |
| CESM LE Therm. Volume Tendency | -0.5 | 0.2 | -1.5 | 0.9 | |
| CESM LE Snowmelt | -0.8 | -1.6 | | | |
| **Freeze Onset** | | | | | |
| PMW Observations | | | | | 6.9 |
| CESM LE Surface Temperature | 5.1 | 6.7 | 1.2 | 8.6 | |
| CESM LE Therm. Volume Tendency | 4.1 | 4.8 | 1.2 | 5.7 | |
| CESM LE Congelation Ice Growth | 4.4 | 5.1 | | | |
| CESM LE Frazil Ice Growth | 3.6 | 4.1 | | | |
| **Melt Season Length** | | | | | |
| PMW Observations | | | | | 10.4 |
| CESM LE Volume – Volume | 4.4 | 4.5 | 1.1 | 6.3 | |
| CESM LE Temperature – Temperature | 3.9 | 5.8 | -0.1 | 7.9 | |
| CESM LE Congelation – Snowmelt | 4.4 | 5.7 | | | |
| CESM LE Frazil – Snowmelt | 3.8 | 4.9 | | | |
| CESM LE Temperature – Snowmelt | 5.6 | 7.1 | | | |

**Table 3.** Trends in pan-Arctic melt onset, freeze onset and melt season length (days/decade) over 1979–2014 using PMW observations and CESM LE definitions.

Relative to the magnitude of the pan-Arctic trends from 1979–2014, the impact of internal variability is very large. For melt onset in the CESM LE, the range of ensemble trends due to internal variability is larger than the magnitude of the melt onset trends. Internal variability even leads to melt onset trends of both signs, even though trends towards earlier melt onset dates





dominate. Freeze onset trends over the satellite era are all positive, but the ensemble spread due to internal variability of 7.4 days/decade is larger than most of the trends in all ensemble members except two (7.5 and 8.6 days per decade, both found using the surface temperature definition).

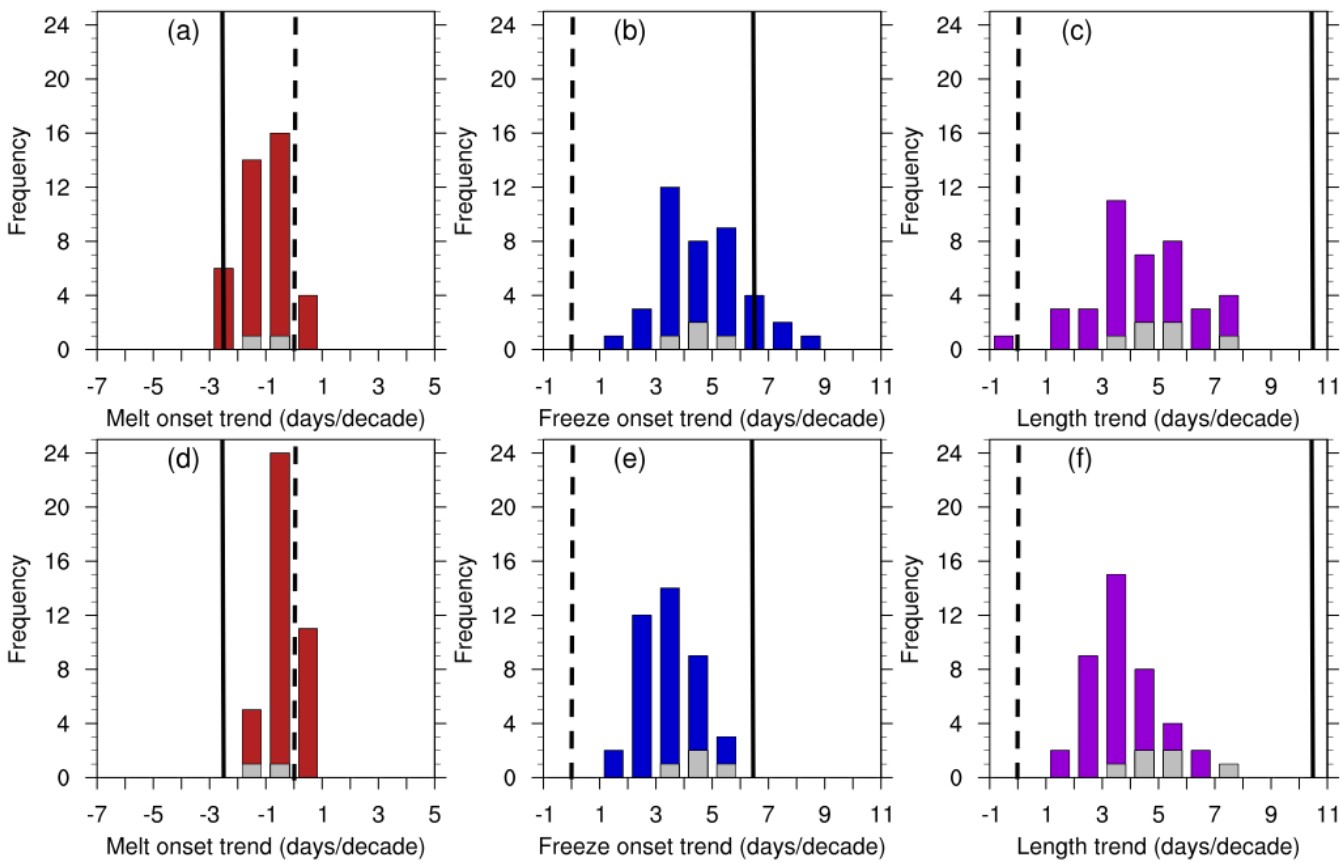

**Figure 6.** Histograms of the trends in pan-Arctic melt season characteristics over 1979–2014 using the surface temperature definitions (a–c) and thermodynamic volume tendency definitions (d–e) for all 40 CESM LE ensemble members. Gray bars represent trends from the other CESM LE definitions for ensemble members 34 and 35. PMW observations are denoted by solid black lines. The zero-line is denoted by dashed black lines. Given the magnitude of the trends, the internal variability is very large. Note that the x-axis limits are different in each panel, but the range is the same (12 days/decade), to facilitate the assessment of the impact of internal variability for different processes and definitions.

Since trends in pan-Arctic freeze onset are consistently larger than melt onset trends, the majority of the trend in melt season
5  length over 1979–2014 stems from the freeze onset component, in agreement with PMW observations (Stroeve et al., 2014). For ensemble members 34 and 35, the Temperature - Snowmelt definition produces the largest trend in melt season length (Table 3). Internal variability in melt season length trends is as large as for the freeze onset trends, with pan-Arctic trends in melt season length between -0.1 and 7.9 days/decade using the surface temperature and thermodynamic ice volume tendency





definitions (Fig. 6). And while the majority of ensemble members show a trend toward a longer pan-Arctic melt season as expected, one member shows a trend toward a shorter melt season over 1979–2014. This demonstrates that internal variability can have a large impact on trends, even over 36-year periods. But by start-year 1981, just two years past the beginning of the satellite period, all ensemble members and definitions have positive 36-year trends in melt season length for the remainder of
the model simulation.

### 3.2.2 Spatial trends

Spatially, trends in melt onset vary differently than trends in freeze onset. Melt onset trends are generally negative except along the Atlantic ice edge, indicating earlier melt onsets across most of the Arctic (Fig. 7). Because the temperature and snowmelt melt onset definitions capture surface processes only, we find that the trends in these definitions are more similar to each other
than to the thermodynamic volume tendency definition, which depends on sea ice melt. In both ensemble members 34 and 35, the snowmelt and surface temperature definitions of melt onset show negative trends in the Laptev, East Siberian and Chukchi Seas that are not present in the thermodynamic ice volume tendency definition, indicating that these trends towards earlier melt represent snow melt, rather than sea ice melt.

CESM LE definitions of freeze onset produce positive trends throughout almost all of the Arctic, indicating later freeze-up,
with the largest trends occurring in marginal ice zones (Fig. 8). The marginal ice zones show the greatest ice loss over the satellite era, and with more open water exposed, trends in sensible and latent heat fluxes have increased (Deser et al., 2000). These fluxes further warm the surface ocean and delay freeze onset. The magnitudes of the freeze onset trends vary between definitions, and there are also regional differences between ensemble members due to internal variability (Fig. 8). However, unlike the trends in melt onset definitions, the regional patterns in freeze onset trends are largely consistent between definitions.
The similarity in trends between definitions based on surface temperature and sea ice variables indicates that temperature trends are driving the delayed freeze-up.

All CESM LE definitions show large positive trends in melt season length in the Barents Sea and in the Laptev and East Siberian Seas, driven by the freeze onset trends in these regions (Fig. 9). Changes in freeze onset are particularly important to changes in the melt season in the marginal ice zones, where sea ice has retreated the most over the satellite period. However,
definition differences and internal variability introduce large variations in the magnitude and even the sign of the diagnosed melt season lengths. The effect of definition differences is most pronounced in the Beaufort Sea, where temperature based definitions indicate a negative trend in melt season length while all other definitions show no or small positive trends in that region (Fig. 9e, j). The effect of internal variability is seen most clearly in the Central Arctic, where even the sign of the trend varies between ensemble members (Fig. 9). Internal variability also affects the magnitude of the melt season trends in the shelf
seas (Fig. 9), as sea ice loss is simulated differently in ensemble member 34 and 35.

### 3.3 Comparing CESM LE and PMW: average melt season characteristics

Pan-Arctic average PMW observations (Stroeve et al., 2014; Markus et al., 2009) fall within the range of model definitions and internal variability for all melt season characteristics (Fig. 1). Spatially, the greatest melt onset similarities exist between the





**Figure 7.** Trends in melt onset dates over 1979–2014 for each CESM LE definition in the two members where they are available (member 34 in a–c, member 35 in d–f) as well as in the PMW satellite observations (g). The snowmelt definition is shown in (a) and (d), the thermodynamic ice volume tendency definition is shown in (b) and (e), and the surface temperature definition is shown in (c) and (f).

CESM LE snowmelt definition and PMW observations, particularly in the central Arctic Ocean and Laptev Sea (Fig. 3). This agrees with the initial expectation that PMW data is most closely related to the snowmelt criteria, as the PMW algorithm is





**Figure 8.** Trends in freeze onset dates over 1979–2014 for each CESM LE definition in the two members where they are available (member 34 in a–d, member 35 in e–h) as well as in the PMW satellite observations (i). The congelation ice growth definition is shown in (a) and (e), the frazil ice growth definition is shown in (b) and (f), the thermodynamic ice volume tendency definition is shown in (c) and (g), and the surface temperature definition is shown in (d) and (h).

designed to detect surface liquid water. Histograms of 1979-2014 average melt onset show that the snowmelt definition agrees best with PMW observations in terms of areal-median and the areal-distribution over the satellite era (Fig. 10). However, the snowmelt definition and PMW observations of average melt onset still do not match exactly. In particular, the snowmelt definition has a greater areal fraction of melt onset dates before June than the PMW data. As both ensemble member 34 and 35 show a similar mismatch, this is likely not due to internal variability, but due to definition differences and/or an early melt onset model bias in the CESM LE.





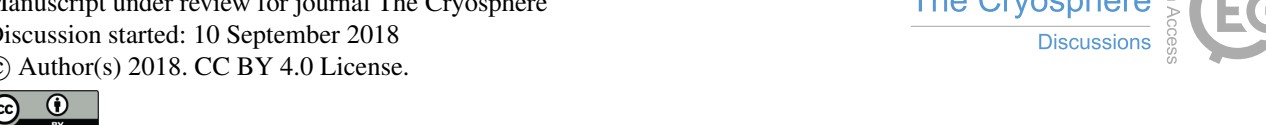

**Figure 9.** Trend in melt season length over 1979–2014 for each CESM LE definition in the two members where they are available (member 34 in a–e, member 35 in f–j) as well as in the PMW satellite observations (k). The Congelation - Snowmelt definition is shown in (a) and (f), the Frazil - Snowmelt definition in (b) and (g), the Volume - Volume definition in (c) and (h), the Temperature - Snowmelt definition in (d) and (j), and the Temperature - Temperature definition in (e) and (j).





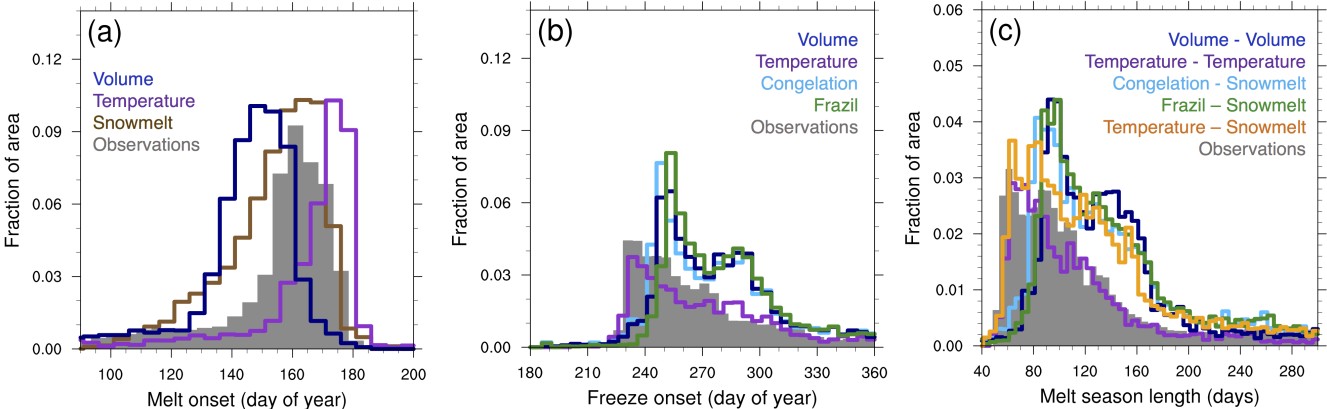

**Figure 10.** Average melt season characteristics from 66°N to 84.5°N for 1979–2014 for PMW satellite observations (filled gray) and each CESM LE definition (in ensemble member 35): (a) melt onset using the surface temperature, thermodynamic ice volume tendency and snowmelt definitions (b) freeze onset using the surface temperature, thermodynamic ice volume tendency, frazil ice growth and congelation ice growth definitions (c) melt season length using the Temperature − Temperature, Temperature − Snowmelt, Volume − Volume, Frazil − Snowmelt and Congelation − Snowmelt definitions. Plots from ensemble member 34 are very similar and are shown in Fig. S.5.

For freeze onset, the surface temperature definition agrees best with PMW observations in terms of median and distribution (Fig. 10). Surface temperature is the only definition for which freeze onset dates in the Central Arctic, Laptev Sea and Kara Seas are not later than PMW observations over the satellite era (Fig. 4). However, all CESM LE freeze onset definitions show dates in the Chukchi Sea roughly 10 days later than in the PMW observations (Fig. 4).

5     Comparisons of melt season length emphasize that no one definition fully captures the PMW observations. All CESM LE definitions show longer melt seasons in the Barents Sea than shown by the PMW data (Fig. 5). By areal fraction, most definitions show a longer melt season length in the CESM compared to PMW data. And while the Temperature - Temperature definition yields a smaller percent area of melt season lengths between 75–125 days than found in PMW observations, the Temperature - Snowmelt definition shows a larger percent area of melt season lengths between 75–125 days than observed (Fig.

10   10). In terms of pan-Arctic averages, CESM LE melt season lengths are both shorter and longer than PMW data depending on definition.

### 3.4   Comparing CESM LE and PMW: trends in melt season characteristics

In the PMW observations spanning 1979–2014 (Markus et al., 2009; Stroeve et al., 2014), pan-Arctic melt onset is occurring 2.5 days earlier per decade and pan-Arctic freeze onset is occurring 6.9 days later per decade (Table 3, Fig. 6). In agreement with

15  PMW data and past studies (Stroeve et al., 2014; Wang et al., 2017), a larger trend in freeze onset than melt onset is produced by all CESM definitions. The PMW melt onset trend falls just outside the range of model definition trends (spanning -2.4 to 0.9 days/decade), while the PMW freeze onset trend is bracketed by model definition trends (spanning 1.2 to 8.6 days/decade).



None of the CESM LE definitions yield trends in melt season length (spanning -0.1 to 7.9 days/decade) as large as the trends found in the PMW observations (Table 3, Fig. 6). In the PMW observations and all but one ensemble member of the CESM LE definitions, the pan-Arctic melt season is lengthening, and this change is driven predominately by later freeze onset dates. But PMW observations show that the average pan-Arctic melt season is lengthening at a rate of 10.4 days per decade,

which is over 30% larger than any of the melt season trends found using CESM LE definitions over the satellite era in any ensemble member (Table 3, Fig. 6). Regionally, we find that the CESM melt season length trends in the marginal ice zones are consistently smaller than the PWM melt season length trends, for all definitions in members 34 and 35 (Fig. 9). In definitions where all 40 ensemble members are available, some members show trends as large satellite observations in certain regions (such as the Barents and Chukchi Seas), but not across the entire marginal ice zone, like what is seen in satellite observations.

This is driven in particular by smaller freeze onset trends in the marginal seas compared to PMW data. All of this suggests that the CESM LE underestimates the melt season length trend, in particular in the marginal seas.

### 3.5 Relationship between melt and freeze onset

Earlier melt and later freeze onset dates are related in both CESM LE definitions and PMW observations (Fig. S.6). In previous work, earlier melt onset has been shown to delay fall freeze onset through increased solar absorption in the Arctic Ocean

(Stroeve et al., 2014). There is moderate correlation between modeled melt and freeze onset in the CESM LE, but there is also substantial internal variability and variations between model definitions. The correlations of melt and freeze onset in the model range between -0.64 and 0.12, while the PMW correlation is -0.26 (Fig. S.6). However, only about 3.5% of all available ensemble members and definitions in the CESM LE show positive correlations, indicating that in general, earlier melt onset dates are related to later freeze onset dates in the same year. This forced relationship between melt onset and freeze onset is

also apparent in the ensemble mean, which shows negative correlation coefficients that bracket the observations (-0.21 using thermodynamic ice volume tendency and -0.49 using surface temperature).

### 3.6 Melt season characteristics and September sea ice

CESM LE members that have the largest trend in September sea ice extent over the period 1979–2014 also have the largest melt season length trend (Fig. 11). Correlations between trends in September sea ice extent and trends in the two CESM

LE melt season length definitions with 40 available ensemble members (surface temperature and thermodynamic ice volume tendency) are both -0.79. In Sec. 3.2 we showed that 36-year trends in melt season characteristics are affected strongly by internal variability. The same is true for September sea ice extent trends, as shown in previous work (Kay et al., 2011; Swart et al., 2015). But unlike the observed trend in melt season length, the observed trend in September sea ice extent falls within the range of internal variability in the CESM LE.

While we cannot discern a bias in CESM LE September sea ice extent trends over the satellite era (Jahn, 2018), a bias may exist for the September sea ice sensitivity (Rosenblum and Eisenman, 2017; Jahn, 2018). An underestimation of melt season length trends could be a contributing factor. Both models and observations have been shown to have an approximately linear relationship between Arctic sea ice extent and global surface temperature (Mahlstein and Knutti, 2012). It has also been found



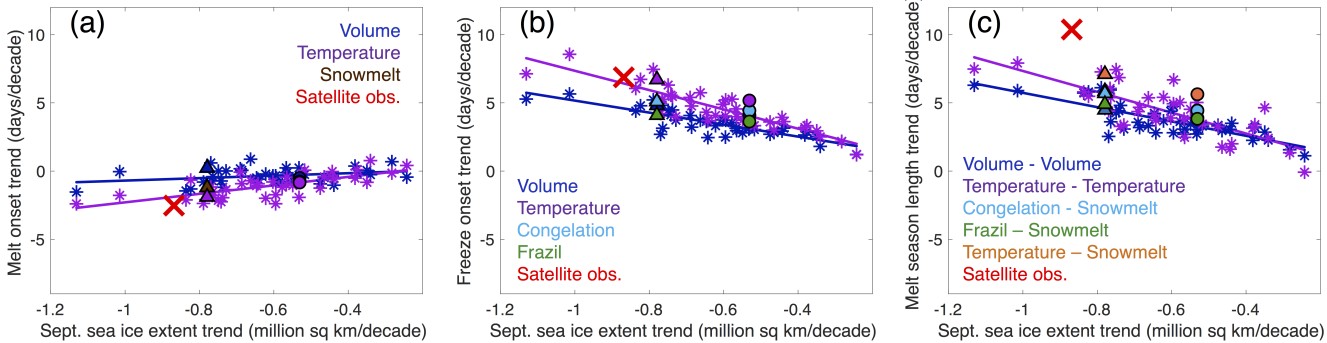

**Figure 11.** Trends in melt season characteristics versus trends in September sea ice extent from 1979–2014 for PMW observations and all available CESM LE ensemble members. Each marker represents an ensemble member. Circles represent ensemble member 34 and triangles represent ensemble member 35. The red markers represent the PMW melt and freeze onset observations and NSIDC September sea ice extent (Fetterer et al., 2002). (a) Trends in melt onset using the surface temperature, thermodynamic ice volume tendency and snowmelt definitions (b) Trends in freeze onset using the surface temperature, thermodynamic ice volume tendency, frazil ice growth and congelation ice growth definitions (c) Trends in melt season length using the Temperature − Temperature, Temperature − Snowmelt, Volume − Volume, Frazil − Snowmelt and Congelation − Snowmelt definitions. Lines represent the least-squares linear fits.

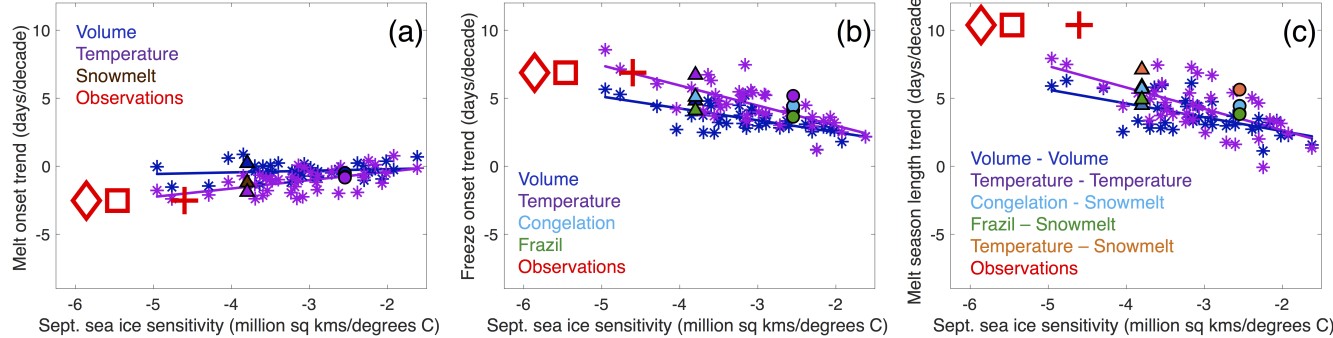

**Figure 12.** Trends in melt season characteristics versus trends in September sea ice sensitivity from 1979–2014 for PMW observations and all available CESM LE ensemble members. Each marker represents an ensemble member. Circles represent ensemble member 34 and triangles represent ensemble member 35. The red markers represent the PMW melt and freeze onset observations and HadCRUT (X), GISTEMP (+), NCDC (diamond) global temperature observations. (a) Trends in melt onset using the surface temperature, thermodynamic ice volume tendency and snowmelt definitions (b) Trends in freeze onset using the surface temperature, thermodynamic ice volume tendency, frazil ice growth, congelation ice growth definitions (c) Trends in melt season length using the Temperature − Temperature, Temperature − Snowmelt, Volume − Volume, Frazil − Snowmelt and Congelation − Snowmelt definitions. Lines represent the least-squares linear fits.

that GCMs producing global warming similar to observations have slower than observed sea ice loss (Rosenblum and Eisenman, 2017). However, large observational uncertainty in sea ice sensitivity (Niederdrenk and Notz, 2018) complicates model





assessment. This agrees with findings for the CESM LE, where the identification of a September sea ice sensitivity bias depends on the selected observations and period (Jahn, 2018). Over the period 1979–2014, September sea ice sensitivity using the GISTEMP (GISTEMP, 2017) global warming trend falls within the ensemble spread, but all ensemble members underestimate the sea ice sensitivity compared to those derived from HadCRUT4 (HadCRUT.4.5.0.0, 2017) and NCDC (NCDC, 2017) global
5  warming trends (Fig. 12). In contrast, all CESM LE ensemble members and definitions underestimate the pan-Arctic trend in melt season length from 1979–2014 (as shown earlier, Fig. 11 and 12). Hence, if the CESM LE is indeed underestimating the September sea ice sensitivity, it is possible that the underestimation of the melt season length trend is a contributing factor.

| Definition Names | 1979–1998 | 2040–2059 | 2080–2099 | 2040–2059 minus 1979–1998 | 2080–2099 minus 1979–1998 |
|---|---|---|---|---|---|
| **Melt Onset** | | | | | |
| Surface Temperature | 160 | 144 | 127 | 16 | 34 |
| Therm. Volume Tendency | 124 | 114 | 96 | 10 | 29 |
| Snowmelt | 155 | 146 | 141 | 9 | 15 |
| **Freeze Onset** | | | | | |
| Surface Temperature | 257 | 319 | 378 | 62 | 120 |
| Therm. Volume Tendency | 272 | 319 | 368 | 47 | 96 |
| Congelation Ice Growth | 273 | 320 | 368 | 48 | 95 |
| Frazil Ice Growth | 278 | 321 | 369 | 43 | 91 |
| **Melt Season Length** | | | | | |
| Temperature - Temperature | 96 | 166 | 245 | 70 | 149 |
| Volume - Volume | 140 | 196 | 268 | 56 | 128 |
| Congelation - Snowmelt | 120 | 174 | 230 | 54 | 111 |
| Frazil - Snowmelt | 122 | 173 | 229 | 51 | 107 |
| Temperature - Snowmelt | 115 | 170 | 226 | 56 | 112 |

**Table 4.** Pan-Arctic ensemble means of melt season characteristics averaged over the time periods 1979–1998, 2040–2059 (mid-century) and 2080–2099 (end of century). Surface temperature and thermodynamic ice volume tendency definitions are averaged over 40 ensembles and all other definitions are averaged over the two ensemble members for which they are available (members 34 and 35).

### 3.7  Pan-Arctic projections under RCP8.5 forcing

All CESM LE definitions project larger changes in freeze onset than in melt onset by the end of the 21st century, and this
10  pattern is consistent with modeled and observed trends over the satellite era. Under RCP8.5 forcing, pan-Arctic melt onset dates are projected to occur 1–2 weeks earlier by the middle of the 20th century while freeze onset dates are projected to occur 1–2 months later (Table 4). By the end of the 21st century, pan-Arctic melt onset dates are projected to occur 2 weeks to a month earlier. At the same time, pan-Arctic freeze onset dates are projected to occur in January of the following year, which is





**Figure 13.** Melt season length averaged over the time periods 1979–1998 (top row), 2040–2059 (middle row) and 2080–2099 (bottom row) using ensemble member 35. Each column is a different definition: (a) Congelation - Snowmelt (b) Frazil - Snowmelt (c) Volume - Volume (d) Temperature - Snowmelt (e) Temperature - Temperature.

3-4 months later than modeled and observed freeze onset dates over the satellite era. Later freeze onset dates are the primary driver of future changes in pan-Arctic melt season length, which is projected to be 5–6 months long by the middle of the 21st century and 7–9 months long by the end of the 21st century (compared to 3-4 months long over the satellite era). The largest changes in projected melt season length are seen in the Chukchi, Beaufort and Barents Seas (Fig. 13).

5    Spatial differences between definitions of melt season length decrease over the 21st century (Fig. 13). This is consistent with the increasing similarity seen in the pan-Arctic means of melt season length (Fig. 1). Variations between definitions decrease as the sea ice extent, and therefore the areal coverage of melt and freeze onset, decreases over the simulation, shrinking the region of study towards the Central Arctic (Fig. S.7). The only definition that gets less similar to the others over time is the snowmelt-derived melt onset definition. This is caused by a more dramatic decrease in areal coverage compared to other melt

10   definitions (Fig. S.7), due to the projected decline of spring snow cover on sea ice (Blanchard-Wrigglesworth et al., 2015).




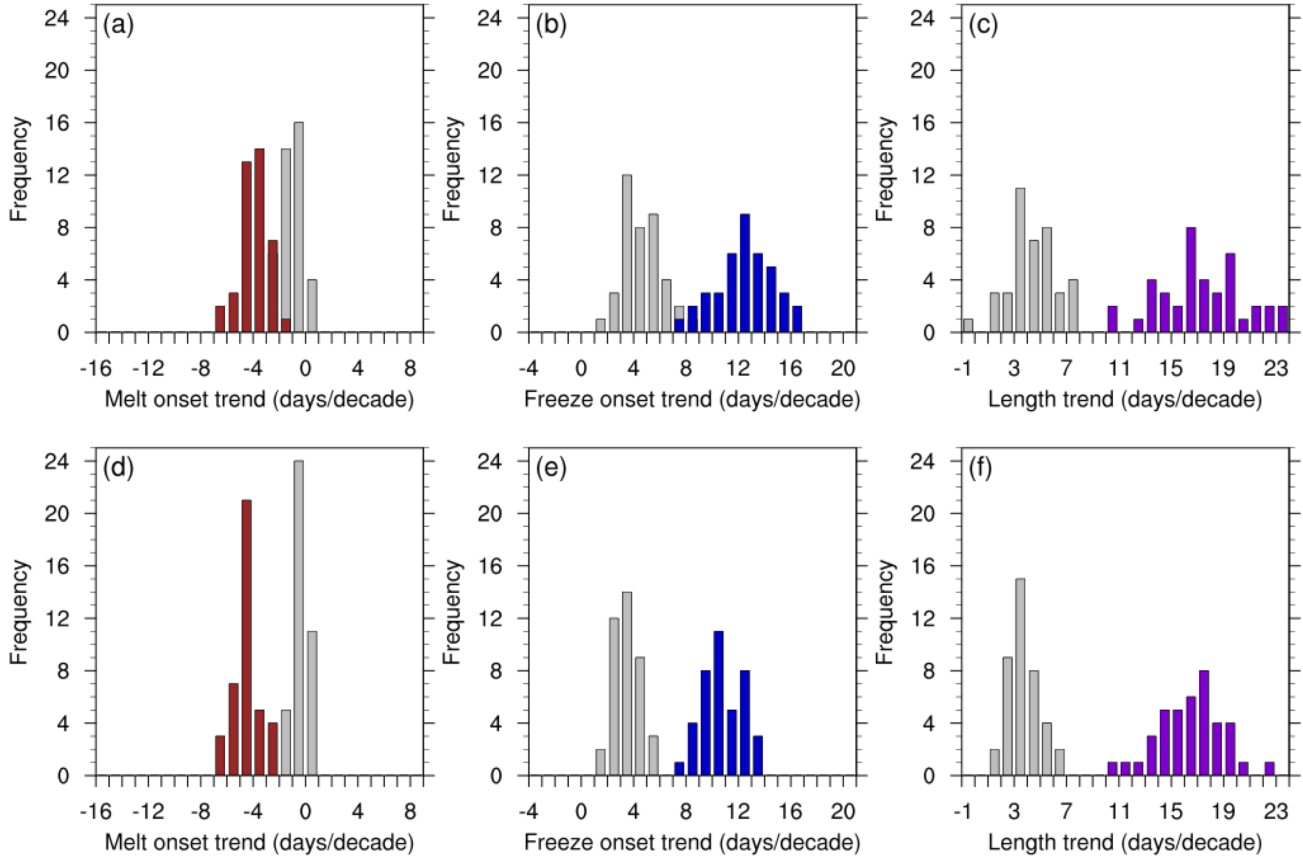

**Figure 14.** Histograms of the pan-Arctic trends in melt season characteristics for 1979–2014 (shaded in gray) and for the end of the 21st century (2064–2099, shaded in red for melt onset, blue for freeze onset and purple for melt season length). This shows the change in the trends over the 21st century as well as the changing impact of internal variability on these trends. The histograms use the surface temperature definitions (first row) and thermodynamic ice volume tendency definitions (second row) for all 40 CESM LE ensemble members. Note that the x-axis range is the same (25 days/decade) for all panels shown in this figure, but different from Fig. 6.

Melt season length definitions become more similar in large part due to the freeze onset component. In particular, the area covered by the surface temperature freeze onset definition becomes more similar to the area covered by the thermodynamic ice volume tendency freeze onset definition (Fig. S.7). This is likely due to the ice growth-thickness relationship (Bitz and Roe, 2004), since thinner ice is less insulating and hence allows freeze onset quickly after temperatures drop below freezing.

5  A lack of insulation also affects the increasingly large area of open water (Barnhart et al., 2016), where changes in surface temperature can quickly trigger frazil ice growth. Thus, as ice coverage decreases, the dates of freeze onset get more similar between surface temperature and thermodynamic ice volume tendency definitions.



Additionally, the internal variability of melt season characteristics depends on definition and is projected to increase through the 21st century. Figures 2 and 6 show that surface temperature definitions of melt onset, freeze onset and melt season length yield greater variations between ensemble members than thermodynamic ice volume tendency definitions over the satellite era. This is also true over the period 2064–2099, as seen in Fig. 14, which shows the shift in ensemble trends between 1979–

2014 and 2064–2099. In all melt season characteristics and definitions, the range of the pan-Arctic trends increases between 1979–2014 and 2064–2099, indicating melt onset, freeze onset and melt season length could be even more affected by internal variability in the future. Average pan-Arctic melt season characteristics also yield greater ranges over 2064–2099 (not shown). Changing internal variability means that future observations will be compared to a wider possible range of modeled melt season characteristics, making model bias detection even more challenging.

**4    Conclusions**

Melt season length plays an important role in the radiation balance of the Arctic and the predictability of sea ice cover. Ideally, we could compare model simulations of melt season characteristics to remote sensing observations to quantify model biases, but there are two major sources of uncertainty in this approach. First, internal variability in the climate system inherently limits how well model projections fit satellite observations of melt and freeze onset (Notz, 2015). Second, there are multiple possible

definitions for sea ice melt and freeze onset in climate models, and none of them exactly correspond to the definitions used by remote sensing methods (Jahn et al., 2012), which rely on PMW brightness temperatures (Markus et al., 2009). In this study, we investigated the impact of definition choices and internal variability for diagnosing Arctic sea ice melt season characteristics (melt onset, freeze onset and melt season length) in model simulation with the CESM LE, with the goals of determining how satellite observations can be used for model evaluation using melt season characteristics and how melt season projections are

impacted by these factors.

We find that while some similarities exist between PMW observations and CESM LE definitions, no single definition fully captures the satellite observations. Definitions of melt season length show impacts of both melt and freeze onset definitions: a large range between definitions, related primarily to the melt onset, and a large range between ensemble members, related primarily to the freeze onset. The average spread between the shortest and longest pan-Arctic melt season length definitions

is over 40 days during the satellite period, primarily because of differences in the melt onset definitions. In particular, the thermodynamic ice volume tendency definition (which is affected by surface, lateral and basal melt) produces melt onset dates much earlier than the surface definitions using snowmelt or surface temperature, which capture snowmelt, rather than ice melt. These results indicate that the choice of melt onset definition is highly dependent on which processes one is aiming to capture–sea ice melt or snowmelt. The PMW observations of melt onset, which capture snowmelt, therefore cannot be used

for comparison to model definitions based on sea ice variables that capture ice melt. Even the snowmelt melt onset definition is not a perfect fit to PMW satellite observations. Furthermore, we find that in the late 21st century, the snowmelt melt onset definition could become less effective for capturing melt onset over large areas of the Arctic, as spring snow cover on sea ice




is projected to decline under RCP8.5 forcing (Blanchard-Wrigglesworth et al., 2015). How this might impact PMW brightness temperature-derived satellite observations is unclear.

In contrast to the melt onset definitions, the investigated freeze onset definitions show greater agreement between each other in terms of both averages, spatial patterns, and trends over the satellite era. However, they are still not identical, as the

surface temperature definition produces slightly earlier freeze onset dates than the other three definitions, which are derived from sea ice variables. The earlier freeze onset dates from the surface temperature definition indicate that changes in surface temperature are driving sea ice formation, therefore producing more comparable definitions for freeze onset than for melt onset (where surface temperature predominantly affects snow melt, but not ice melt). Furthermore, future projections show that the CESM LE definitions of freeze onset become even more similar to each other over time. This is likely due to thinning ice,

which reduces insulation and allows for faster ice growth once surface temperatures fall below freezing (Bitz and Roe, 2004). The fact that surface temperature drives ice growth also has important implications for internal variability. CESM LE freeze onset definitions experiences greater internal variability than melt onset definitions. Similarly, surface temperature definitions are more variable than those based on thermodynamic ice volume tendency. This shows that the internal variability of a selected definition variable impacts the internal variability of the derived melt and freeze onset.

In both PMW observations and CESM LE definitions, early pan-Arctic melt onset tends to be followed by later pan-Arctic freeze onset over the satellite era, in agreement with previous work (Stroeve et al., 2014). However, while the ensemble mean clearly shows this forced response, internal variability affects this relationship and can reverse this relationship for individual years in the CESM LE over the satellite era.

The pan-Arctic trend in melt season length is driven mostly by the trend in freeze onset in the CESM LE, in agreement with

previous work for the PMW melt season length (Stroeve et al., 2014). Yet, despite the use of multiple plausible definitions and 40 ensemble members, no model definition produce trends in the pan-Arctic melt season as large as PMW observations. The inability of the CESM to produce pan-Arctic melt season lengths as large as observations suggests a model bias. In particular, the marginal ice zones consistently show smaller trends for all model definitions of freeze onset and melt season length than PMW observations. This melt season trend bias may have important implications for September sea ice. High correlations exist

between September sea ice sensitivity and melt season length over the satellite era in the CESM LE. Observational uncertainty in sea ice sensitivity is substantial (Niederdrenk and Notz, 2018), but the data used here indicate that the CESM LE may underestimate September sea ice sensitivity. It is therefore possible that an underestimation of the trend in CESM LE melt season length is one factor contributing to the potential biases in the simulated sea sensitivity in the CESM.

Under RCP8.5 forcing, the CESM LE projects that the Arctic sea ice melt season will last 7–9 months by the end of the

21st century, compared to 3–4 months over the satellite era, with later freeze onset dates continuing to be the dominant driver of these changes. Internal variability in melt season characteristics is also projected to increase by the end of the 21st century. This means that definition differences and internal variability will continue to be factors complicating model-observation comparisons of the Arctic sea ice melt season, particularly since they are both projected to change over time.



*Data availability.* CESM LE data are publicly available at the National Center for Atmospheric Research Climate Data Gateway (https://www.earthsystemgrid.org/). PMW satellite observations are publicly available at the NASA Cryosphere Science Research Portal (https://neptune.gsfc.nasa.gov/csb/).

*Author contributions.* AJ conceived the study, AA analyzed the data and prepared the manuscript, with guidance and edits from AJ.

5 *Competing interests.* No competing interests are present.

*Acknowledgements.* This work is supported by the National Science Foundation Graduate Research Fellowship under Grant No. DGE 1144083. We thank Drs. Julienne Stroeve and Thorsten Markus for sharing their passive microwave dataset of melt and freeze onset dates and helpful discussions. Discussions with Drs. David Bailey, Julienne Stroeve, Maxwell Boykoff and Jennifer Kay on this work and discussions at the 2016 FAMOS workshop are appreciated. The CESM project is supported by the National Science Foundation and the Office of 10 Science (BER) of the U.S. Department of Energy. Computing resources for the CESM ensembles were provided by the Climate Simulation Laboratory at NCAR's Computational and Information Systems Laboratory (CISL), sponsored by the National Science Foundation and other agencies. Five of the CESM LE simulations were produced at the University of Toronto under the supervision of Dr. Paul Kushner. NCL (NCL, 2017) was used for data analysis.



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
