# Peer review of "Definition differences and internal variability affect the simulated Arctic sea ice melt season"

_The Cryosphere, 2018_

## Referee Comment (RC1) · Anonymous Referee #1 · 27 Sep 2018

This work investigates the dates of sea ice melt onset, freeze onset, and the length of the melt season as identified from a variety of daily CESM LE model outputs. The authors examine the melt season by defining several melt and freeze onset estimates from the model output variables that make physical sense to replicate melt and freeze onset dates derived from passive microwave satellite observations, highlighting the mismatch between melting parameters available from the models and melt as observed from brightness temperatures. The analysis provides a through comparison of several definitions of melt from the models and assesses the effect of internal variability on the melt season definitions.

[Figure]

Results show that none of the model definitions produce trends in melt season length as large as those from the satellite data and that the melt onset dates are less consistent across definitions than freeze onset dates. Further, some of the model definitions are similar to the satellite observations, but no single definition replicates the satellite observations.

The paper is generally well written and reports an interesting study of some of the challenges when trying to compare model output with observations. Overall, I think the study is timely, thorough, and of value for both the modeling and remote sensing communities. Before publication I recommend that the authors check that their data are accurately cited and clarify a few points in their data and methodology section as noted in my specific comments.

Specific Comments

P3, L4: The AHRA method for MO detection is independent from the Smith (1998) method for MO, while the Markus PMW algorithm incorporates the Smith method as one of its components. In line 4, it is more accurate to state that the AHRA method to detect MO dates is an improvement over earlier work that provided melt onset dates only over MYI rather than stating that the AHRA "builds upon" the Smith method.

P3, L11: "All of this..." is too vague. I suggest modifying this sentence to begin by stating what this is referring to.

P4, L5: The Markus data set gets updated and modified periodically. Please cite where and specifically when the data were accessed in this section.

P4, L14: It would be worth noting the continuous melt season length as defined here differs from various melt season lengths defined in other studies of PMW melt/freeze dates. For example, Stroeve et al., (2014) defines inner (EFO-CMO) and outer (CFO-EMO) melt season lengths that differ from the CFO-CMO melt season used here.

Table 1: Assuming that the threshold values are met for the appropriate number of

consecutive days shown in this table, is the initial day of the 3, 5, or 21 day consecutive timescale accepted as the MO or FO date? It wasn't clear to me from reading the text (page 5, lines 3-11) and the supplement which date in the time period is used as the onset date.

Figure 3: I suggest citing the Bootstrap sea ice concentration data in the figure caption as "(Comiso, 2017)" and in the list of references as requested by NSIDC: Comiso, J. C. 2017. Bootstrap Sea Ice Concentrations from Nimbus-7 SMMR and DMSP SSM/I-SSMIS, Version 3. [Indicate subset used]. Boulder, Colorado USA. NASA National Snow and Ice Data Center Distributed Active Archive Center. doi: https://doi.org/10.5067/7Q8HCCWS4I0R. [Date Accessed].

In the result sections, the place names used are fairly common (e.g., Central Arctic, East Siberian Sea, etc.), but may not be known to all readers. Labels could be added to one of the panels in Figure 3 or a region map identifying the locations referred to could be placed in the supplement.

P14, L3-5: This sentence is a bit awkward. Are you stating that adjusting the 36-year time series window to begin in 1981 removes the outlier positive trend in Figure 6c? I suggest rephrasing and not beginning the sentence with but.

P14, L26: Specify that the largest differences are to the north of the Beaufort Sea rather than in the Beaufort Sea.

P14, L29-30: Spell out what you mean by "shelf seas" since this term isn't used elsewhere in the paper.

Figure 7: This comment really applies to all of the maps, but is most apparent in figure 7. How are the effects of the retreating ice edge in the Atlantic sector during the satellite era accounted for in the calculation of the statistics shown in your maps? Are nans ignored? Could this be contributing to the mix of positive and negative trends seen in the Barents Sea and east of Greenland? A statement (a few sentences) about how

the statistics are calculated should be included in the methodology section and some discussion in text of how changes in the ice edge may be contributing to the patterns shown in Fig 7a-f could be included near P14, L8.

Figure 11: Cite the sea ice index data with the following reference: Fetterer, F., K. Knowles, W. Meier, M. Savoie, and A. K. Windnagel. 2017, updated daily. Sea Ice Index, Version 3. [Indicate subset used]. Boulder, Colorado USA. NSIDC: National Snow and Ice Data Center. doi:https://doi.org/10.7265/N5K072F8. [Date Accessed].

P19, L30: A short description of how sea ice sensitivity is derived should be included.

Technical Corrections

P4, L7-8: The acronym for scanning multichannel microwave radiometer should be SMMR.

P9, L4: I suggest changing large gradients to strong gradients or tight gradients.

P9, L6: I also suggest changing "Even larger gradients..." to Stronger gradients.

P14, L1: I suggest beginning the sentence as "While the majority..."

P19, L7: Typo - PMW

P25, L28: Should this be "sea ice sensitivity"?

---

## Referee Comment (RC2) · Anonymous Referee #2 · 28 Sep 2018

General comments

The paper investigates the difference in model definitions of melt onset and freeze-up using Community Earth System Model large Ensemble (CESM LE) and compared the results to the observation datasets from passive microwave (PMW) sensors. With 35 years of PMW observed melt/freeze dates were compared with different model outputs from CESM LE using different definitions based on different sensible physical processes. In addition, melt season lengths were calculated by combining varied definitions. The study concluded that none of the model outputs of melt/freeze-up dates matches with PMW results. The variation in melt dates is prominent compared to

freeze-up dates. The authors argued that these variations are mainly due to the varied surface/bottom processes involved during melt timing. In case of melt season length, the trend found in PMW observation in different from model simulations.

The paper is well written in a logical manner with necessary details. Figures are clear and well presented. The supplementary materials also complement the paper with additional information. The discussion was easy to follow. I have some major concerns with the content of the manuscript that need to be addressed.

The timing of melt from ice volume definition is unexpected. Fig 1a (and Fig 2d) shows MO date during YD (day of the year) 120-125, based on volume tendency definition. Most of the Arctic Ocean generally remains cold, therefore cannot initiate surface melt during that time of the year. For bottom melt, the ocean is still not warm enough during YD 120-130 to initiate bottom melt. Thermodynamically, none of the processes supports ice melt during this period. Considering the warm Atlantic/Pacific water intrusion in the Arctic, I think it should not result in basal melt all over the Arctic. Fig 3b shows large spatial differences in MO, where the Atlantic part of the Arctic experienced very early basal melt, which is expected. Perovich et al. (2008, GRL) observed bottom melt in the Beaufort Sea around YD 150. I would expect more variation of MO from ice volume tendency definition in Fig 1a, where the spread could be more towards YD 150-160. A descriptive statistics (number of grids vs MO date for ice volume tendency) would help to understand the variation in MO date in the region. A substantial effort is made to compare the model results with PMW observations to detect melt/freeze up while considering the detection of melt/freeze from PMW observation is absolute truth. Therefore, I strongly believe there should be a section describing PMW observation techniques to detect melt and freeze in a concise manner. This will help the reader to understand the process considered in the detection of melt/freeze up using PMW observations. The detection errors/limitation (from PMW observation) should be taken into consideration while comparing the results with model output. The melt/freeze timing difference between models and PMW observation could result due to multi-sensor

calibration issues including detection methods of state variables, rather than definition diversity. Authors should consider this aspect in the discussion. This manuscript can be accepted after addressing these major concerns properly. I also provide some minor comments afterwards. I suggest Major revision.

Specific comments

Page 2 Line 1: The timing of ice retreat not necessarily defines melt onset (MO). After MO, other thermodynamic regimes (e.g. pond onset, drainage, break up) are observed in the Arctic before the ice starts to retreat. MO is a function on ice/snowmelt on sea ice, which can be detected by both passive and active microwave sensors, which is not the same as ice retreat.

Page 3 line 10: ". . . but large difference. . .." Is the mean difference statistically significant?

Page 4 line 25: ". . . a second using surface temperature. . ." is it NSTM or daily mean?

Page 4, line 29: most of the sea ice in the Arctic is found to be snow covered. As result, the ice melt would place much later in the season compared to snowmelt onset. Mostly, during pond onset, which is generated from snowmelt water, standing liquid water on ice surface starts melting the ice surface. Moreover, most of the ice melt takes place when the pond is drained and ice surface is exposed directly to the atmosphere. This ice-melt definition seems unrealistic in real-world scenarios in the Arctic.

Page 7 Fig 1a: Snowmelt and temperature definition has a good agreement until 2045. After that, snowmelt tends to occur well before temperature hits -1C. What physical process might cause this? Any reasonable explanation?

Page 9 Fig 3C: Why the Canadian Arctic Archipelago is not displayed in the MO map?

Page 10 Fig 4: Looks like all model definitions found Arctic warmer than it should be that ultimately delaying the freeze up compared to the PMW observations. This pattern is prominent, especially for the Central Arctic Ocean. It is interesting to see all

definitions captured the spatial variability at the MIZ.

---

## Author Comment (AC1) · 3 Dec 2018

Response to Referee 1

*This work investigates the dates of sea ice melt onset, freeze onset, and the length of the melt season as identified from a variety of daily CESM LE model outputs. The authors examine the melt season by defining several melt and freeze onset estimates from the model output variables that make physical sense to replicate melt and freeze onset dates derived from passive microwave satellite observations, highlighting the mismatch between melting parameters available from the models and melt as observed*

[Figure]

*from brightness temperatures. The analysis provides a through comparison of several definitions of melt from the models and assesses the effect of internal variability on the melt season definitions.*

*Results show that none of the model definitions produce trends in melt season length as large as those from the satellite data and that the melt onset dates are less consistent across definitions than freeze onset dates. Further, some of the model definitions are similar to the satellite observations, but no single definition replicates the satellite observations.*

*The paper is generally well written and reports an interesting study of some of the challenges when trying to compare model output with observations. Overall, I think the study is timely, thorough, and of value for both the modeling and remote sensing communities. Before publication I recommend that the authors check that their data are accurately cited and clarify a few points in their data and methodology section as noted in my specific comments.*

**We thank the referee for his/her thoughtful and constructive comments. We have revised the data citations and clarified the issues raised. Each specific comment is addressed below individually. The original referee comments are shown in italic, the author reply in bold.**

**Specific comments:**

*P3, L4: The AHRA method for MO detection is independent from the Smith (1998) method for MO, while the Markus PMW algorithm incorporates the Smith method as one of its components. In line 4, it is more accurate to state that the AHRA method to detect MO dates is an improvement over earlier work that provided melt onset dates only over MYI rather than stating that the AHRA "builds upon" the Smith method.*

**Action taken: Thank you for the clarification. The phrasing has been changed as suggested.**

*P3, L11: "All of this..." is too vague. I suggest modifying this sentence to begin by stating what this is referring to.*

**Action taken: We agree. The sentence has been rephrased for specificity.**

*P4, L5: The Markus data set gets updated and modified periodically. Please cite where and specifically when the data were accessed in this section.*

**Action taken: The data are now cited with information about how and when the data were accessed.**

*P4, L14: It would be worth noting the continuous melt season length as defined here differs from various melt season lengths defined in other studies of PMW melt/freeze dates. For example, Stroeve et al., (2014) defines inner (EFO-CMO) and outer (CFO-EMO) melt season lengths that differ from the CFO-CMO melt season used here.*

**Action taken: A note has been made to distinguish the melt season length definition used here from the melt season lengths used in Stroeve et al., (2014).**

*Table 1: Assuming that the threshold values are met for the appropriate number of consecutive days shown in this table, is the initial day of the 3, 5, or 21 day consecutive timescale accepted as the MO or FO date? It wasn't clear to me from reading the text (page 5, lines 3-11) and the supplement which date in the time period is used as the onset date.*

**Action taken: The initial day of the timescale is the accepted melt onset/freeze onset date. A statement has been added to the Supplementary Material to clarify this.**

*Figure 3: I suggest citing the Bootstrap sea ice concentration data in the figure caption as "(Comiso, 2017)" and in the list of references as requested by NSIDC: Comiso, J. C. 2017. Bootstrap Sea Ice Concentrations from Nimbus-7 SMMR and DMSP SSM/I-SSMIS, Version 3. [Indicate subset used]. Boulder, Colorado USA.NASA National Snow and Ice Data Center Distributed Active Archive Center.doi:https://doi.org/10.5067/7Q8HCCWS4I0R. [Date Accessed]. In the result sections, the place names used are fairly common (e.g., Central Arctic, East Siberian Sea, etc.), but may not be known to all readers. Labels could be added to one of the panels in Figure 3 or a region map identifying the locations referred to could be placed in the supplement.*

**Action taken: Comiso, (2017) is now cited with the appropriate information. A map including labels of most major features of the Arctic Ocean and the marginal seas has been included in the Supplementary Material and referenced in the Methods Section.**

*P14, L3-5: This sentence is a bit awkward. Are you stating that adjusting the 36-year time series window to begin in 1981 removes the outlier positive trend in Figure 6c? I suggest rephrasing and not beginning the sentence with but.*

**Author response: Adjusting the 36-year time series window to begin in 1981 removes the outlier positive trend in Figure 6c. This is true for all 36-year trend start-years after 1981 as well.**

**Action taken: The statement has been rephrased for clarity.**

*P14, L26: Specify that the largest differences are to the north of the Beaufort Sea rather than in the Beaufort Sea.*

**Action taken: As suggested, we now specify that the largest differences are to the north of the Beaufort Sea rather than in the Beaufort Sea.**

*P14, L29-30: Spell out what you mean by "shelf seas" since this term isn't used elsewhere in the paper.*

**Action taken: Changed phrasing of "shelf" seas to "marginal" seas for consistency with other uses in the paper.**

*Figure 7: This comment really applies to all of the maps, but is most apparent in figure 7. How are the effects of the retreating ice edge in the Atlantic sector during the satellite era accounted for in the calculation of the statistics shown in your maps? Are nans ignored? Could this be contributing to the mix of positive and negative trends seen in the Barents Sea and east of Greenland? A statement (a few sentences) about how the statistics are calculated should be included in the methodology section and some discussion in text of how changes in the ice edge may be contributing to the patterns shown in Fig 7a-f could be included near P14, L8.*

**Author response: Trends in melt season characteristics at individual grid cells are calculated as the slope of the least squares linear regression from 1979–2014, given that the grid cell has a valid melt season characteristic for at least 20 years of the 36-year period. NaNs are ignored, and trends are not calculated for grid cells with less than 20 years of melt/freeze/melt season length data. This is the same approach used in Stroeve et al., (2014). The changing ice edge is therefore only partially accounted for and could still affect the reliability of the trends along the ice edge.**

**Action taken: A description of the methods used for pan-Arctic means, pan-Arctic trends, and individual grid cell trends has been included in the Methods section. We have also added the mean position of the ice edge to Figure 7 as in Figure 3, and included some additional discussion of the impact of the changing Atlantic ice edge near the recommended location in the original manuscript.**

*Figure 11: Cite the sea ice index data with the following reference: Fetterer, F., K. Knowles, W. Meier, M. Savoie, and A. K. Windnagel. 2017, updated daily. Sea Ice Index, Version 3. [Indicate subset used]. Boulder, Colorado USA. NSIDC: National Snow and Ice Data Center. doi:https://doi.org/10.7265/N5K072F8. [Date Accessed].*

**Action taken: Fetterer et al., (2017) is now cited with the appropriate information.**

*P19, L30: A short description of how sea ice sensitivity is derived should be included.*

**Action taken: As suggested, a short description of how sea ice sensitivity is derived has been included.**

**All suggested technical corrections from Referee 1 were implemented.**

---

## Author Comment (AC2) · 3 Dec 2018

Response to Referee 2

*The paper investigates the difference in model definitions of melt onset and freeze-up using Community Earth System Model large Ensemble (CESM LE) and compared the results to the observation datasets from passive microwave (PMW) sensors. With 35 years of PMW observed melt/freeze dates were compared with different model outputs from CESM LE using different definitions based on different sensible physical processes. In addition, melt season lengths were calculated by combining varied def-*

*initions. The study concluded that none of the model outputs of melt/freeze-up dates matches with PMW results. The variation in melt dates is prominent compared to freeze-up dates. The authors argued that these variations are mainly due to the varied surface/bottom processes involved during melt timing. In case of melt season length, the trend found in PMW observation in different from model simulations.*

*The paper is well written in a logical manner with necessary details. Figures are clear and well presented. The supplementary materials also complement the paper with additional information. The discussion was easy to follow. I have some major concerns with the content of the manuscript that need to be addressed.*

**We thank the referee for his/her thoughtful and constructive comments. Each comment/concern is addressed below individually. The original referee comments are shown in italic, the author replies and actions taken are in bold. Figures R.1.–R.3. are found in the supplementary file tc-2018-183-supplement.pdf.**

**General comment: Part 1**

*The timing of melt from ice volume definition is unexpected. Fig 1a (and Fig 2d) shows MO date during YD (day of the year) 120-125, based on volume tendency definition. Most of the Arctic Ocean generally remains cold, therefore cannot initiate surface melt during that time of the year. For bottom melt, the ocean is still not warm enough during YD 120-130 to initiate bottom melt. Thermodynamically, none of the processes supports ice melt during this period. Considering the warm Atlantic/Pacific water intrusion in the Arctic, I think it should not result in basal melt all over the Arctic. Fig 3b shows large spatial differences in MO, where the Atlantic part of the Arctic experienced very early basal melt, which is expected. Perovich et al. (2008, GRL) observed bottom melt in the Beaufort Sea around YD 150. I would expect more variation of MO from ice volume tendency definition in Fig 1a, where the spread could be more towards YD 150-160. A descriptive statistics (number of grids vs MO date for ice volume tendency)*

*would help to understand the variation in MO date in the region.*

**Author response: Through Figures 1a and 3b we are aiming to show the impacts of warm Atlantic inflow on melt onset dates derived from an ice melt definition (thermodynamic ice volume tendency). Figure R.1. shows that in the CESM LE, the largest spring basal melt rates are found in the Atlantic inflow region, as expected by the referee. In the CESM LE, basal melt occurs in the Central Arctic later in the summer (not shown). Average melt onset dates from the thermodynamic ice volume tendency definition over the satellite era (shown in Figure 3b) are earlier in the inflow regions than those derived from surface definitions in the CESM LE (snowmelt, surface temperature) and satellite observations, since the thermodynamic ice volume tendency definition captures the early spring basal melt in the inflow regions. Figure R.2. shows an example of each variable's evolution leading up to melt onset from a grid cell just north of Svalbard for the year 1979, where the melt onset date from thermodynamic ice volume tendency occurs before the melt onset dates from the snowmelt and surface temperature definitions.**

**In the manuscript, we argue that the earlier melt onset dates in the inflow regions found using the thermodynamic ice volume tendency definition are related to warm Atlantic water triggering bottom melt that would not be captured in the surface definitions or satellite observations. These earlier melt onset dates lower the pan-Arctic mean of the thermodynamic ice volume tendency definition (as seen in Figure 1a, which shows pan-Arctic means of each definition over the area 66-84.5 N).**

**Action taken: We have replaced Figure 3 in the original manuscript with Figure R.3. Figure R.3. is identical to Figure 3, but Figure R.3. has an extended scale to clearly demonstrate the range of melt onset dates in the Atlantic inflow region, and the large differences between the different definitions in that region. We have also added a discussion of the very early melt onset in the inflow regions, now clearly shown in the revised Figure R.3., and how it affects the pan-Arctic thermodynamic ice volume tendency definition of melt onset.**

**General comment: Part 2**

*A substantial effort is made to compare the model results with PMW observations to detect melt/freeze up while considering the detection of melt/freeze from PMW observation is absolute truth. Therefore, I strongly believe there should be a section describing PMW observation techniques to detect melt and freeze in a concise manner. This will help the reader to understand the process considered in the detection of melt/freeze up using PMW observations. The detection errors/limitation (from PMW observation) should be taken into consideration while comparing the results with model output. The melt/freeze timing difference between models and PMW observation could result due to multi-sensor calibration issues including detection methods of state variables, rather than definition diversity.*

**Author response:** We agree that a section on the details of the PMW observation techniques is useful for the reader, so they do not have to refer to Markus et al., (2009). We also very much agree that observational uncertainty is an important aspect of model evaluation and believe that it should be addressed where possible. In this case, the satellite data is not provided with error bars, and the differences between inter-calibration methods have only been studied for melt onset dates (Bliss et al., 2017) and not for freeze onset dates. We therefore have to rely on the Bliss et al., 2017 comparison of melt onset data sets for guidance on how the CESM LE definitions may be evaluated while considering observational uncertainty between the PMW and AHRA algorithms. There are limitations to this, since Bliss et al., (2017) used a combined version of PMW data that incorporates both early and continuous melt onset dates (called Passive Microwave Combined Melt Onset or PMWC), while we use only continuous melt onset dates so that we can determine continuous melt season length. Thus, many of the conclusions we might draw about the effects of observational uncertainty are incomplete or not fully representative of the data used in our study.

**Action taken:** We have changed the way that the satellite observations are introduced to the manuscript, to allow for a more detailed description of the methodology. Most of the description of satellite observations of melt and freeze onset that was originally found in the Introduction has been moved the the Methods section, where the PMW method is now described in more detail and in the context of comparing observations to GCMs.

A statement has also been added to Section 3.3 on how the observational uncertainty between AHRA melt onset dates and PMW Combined (PMWC) melt onset dates could influence diagnoses of model biases using the results of the study Bliss et al., (2017). We additionally discuss how Bliss et al., 2017 compared early melt onset algorithms while we assess continuous melt onset, and how it is un-

**clear if the observational uncertainty is the same for early and continuous melt onset.**

**Specific comments:**

*Page 2 line 1: The timing of ice retreat not necessarily defines melt onset (MO). After MO, other thermodynamic regimes (e.g. pond onset, drainage, break up) are observed in the Arctic before the ice starts to retreat. MO is a function on ice/snowmelt on sea ice, which can be detected by both passive and active microwave sensors, which is not the same as ice retreat.*

**Action taken: We agree and have rephrased this section so that melt onset and ice retreat are not construed as the same process.**

*Page 3 line 10: "...but large difference..." Is the mean difference statistically significant?*

**Author response: In Bliss et al., (2017) the mean differences between PMWC and AHRA are calculated regionally and for the total Arctic. The statistical significance of each mean difference is not assessed. The slopes of the PMWC and AHRA time series in each area are evaluated using a Student's T-test at the 95% confidence level to determine whether or not they are equal.**

**Action taken: This sentence has been rephrased to remove discussion of statistical significance for both mean differences and trends so that they do not appear to contrast.**

*Page 4 line 25: "...a second using surface temperature.." is it NSTM or daily mean?*

**Action taken: Daily mean surface temperature is used and this has been clarified in the text.**

*Page 4 line 29: most of the sea ice in the Arctic is found to be snow covered. As result, the ice melt would place much later in the season compared to snowmelt onset. Mostly, during pond onset, which is generated from snowmelt water, standing liquid water on ice surface starts melting the ice surface. Moreover, most of the ice melt takes place when the pond is drained and ice surface is exposed directly to the atmosphere. This ice-melt definition seems unrealistic in real-world scenarios in the Arctic.*

**Author response: The melt onset definition derived from thermodynamic ice volume tendency is meant to act as one of multiple plausible definitions of melt onset. Thermodynamic ice volume tendency in the CESM LE includes surface, lateral and bottom ice melt.**

**While the resulting melt onset from thermodynamic ice volume tendency is dissimilar to the snowmelt and surface temperature-based definitions of melt onset, it demonstrates how the selection of a melt onset definition is dependent on the question posed (especially when comparing to satellite observations). It also provides insight into the timing of specific processes (snowmelt versus ice melt) at the beginning of the melt season. It is useful to assess the differences in timing between snow melt and ice melt, since snow versus ice melt matters for certain applications (e.g., biogeophysical processes in the ice, sediment and contaminant transport by sea ice), but PMW data can only provide information about snow melt (not basal or lateral melt).**

**Action taken: Along the lines of the author response above, a statement has been added to the Methods sections clarifying the rationale for using a variety of melt onset definitions and the possible applications of these definitions.**

*Page 7 Fig 1a: Snowmelt and temperature definition has a good agreement until 2045. After that, snowmelt tends to occur well before temperature hits -1 C. What physical process might cause this? Any reasonable explanation?*

**Author response: In Section 3.7 (pg. 22 lines 5-10) of the original manuscript the others note that the snowmelt-derived melt onset definition has a more dramatic decrease in areal coverage compared to the other melt definitions (Fig. S.7), due to the projected decline of spring snow cover on sea ice (Blanchard-Wrigglesworth et al., 2015). The larger decline in areal coverage of the snowmelt definition likely limits our ability to effectively compare it to the surface temperature definition after 2045 in Figure 1a, which shows pan-Arctic means, and this is discussed in Section 3.7.**

*Page 9 Fig 3C: Why the Canadian Arctic Archipelago is not displayed in the MO map?*

**Author response: The variable "daily mean surface temperature" is used to derive melt and freeze onset and is output on the atmospheric model component grid (0.9x1.25), while the other variables used are output on the sea ice model component grid (gx1v6). We use surface temperature regridded to the CICE grid only to calculate the freeze onset dates in the Temperature - Snowmelt melt season length definition (daily mean snowmelt is only available for ensemble members 34 and 35 and on the gx1v6 grid). Otherwise, surface temperature is used on its original grid to minimize errors associated with regridding. This is noticeable in Figure 3C, where the Canadian Arctic Archipelago is not resolved in the atmospheric grid (0.9x1.25).**

**Action taken: A statement has been added to the Methods section about grid differences between variables and when regridding takes place.**

*Page 10 Fig 4: Looks like all model definitions found Arctic warmer than it should be that ultimately delaying the freeze up compared to the PMW observations. This pattern is prominent, especially for the Central Arctic Ocean. It is interesting to see all definitions captured the spatial variability at the MIZ.*

**Author response: Both Figure 4 and Figure 10 in the original manuscript show that the surface temperature definition of freeze onset agrees well with PMW observations, while the other definitions produce a higher areal fraction of later freeze onset dates, which is found particularly in the central Arctic. It is expected that the conditions for freeze onset should be met earlier in the surface temperature definition than in the definitions based on ice growth, due to the delay between temperature reaching the freezing point and actual ice formation. This good agreement between PMW freeze onset and the temperature definition of freeze onset also agrees well with findings from Markus et al., (2009). In Markus et al., (2009), PMW freeze onset dates agreed very well with an observational surface air temperature definition based on POLES buoy data. The authors wrote, "For the central Arctic, PMW freezeup dates agree best with the POLES continuous freeze days" (Markus et al., 2009).**

**It is likely that the PMW agrees well with surface temperature and surface air temperatures definitions since these represent strictly surface processes. However, refreezing of ponds or liquid water in the snow on sea ice is not accounted for in the CESM. Hence, the ice-based definitions of freeze onset in the model do not reflect such surface processes, which do occur in reality, in particular in the central Arctic where there are currently high ice concentrations throughout the summer. It therefore makes sense that in the Central Arctic, the PMW data shows an earlier freeze onset than the CESM does for ice-based definitions, as the PMW data likely captures surface/snow processes, which are only captured by the surface temperature definition in the model, and would lead to surface refreezing if such processes were to be simulated. To come back to the referee's comment about a warmer Arctic, we do not think that the later freeze onset in the CESM compared to PMW reflects a too warm central Arctic Ocean, but rather points to the impact of definition differences and lack of model processes.**

[Figure]

**Action taken: A statement on the impacts of surface snow refreezing versus ice-variable based freeze onset and the lack of model physics in this regard has been added to the Methods section of the revised manuscript. Discussion on how this affects comparison with PMW observations has also been added to the Results section and Conclusions.**

**Supplement:**

**Figures for " Response to Referee 2"**

[Figure]

Figure R.1.: Average daily basal melt rate over April, May and June from 1979–2014 in the CESM LE using ensemble member 35.

[Figure]

Figure R.2.: Each CESM LE melt onset definition variable over days 75-225 of year 1979 in a grid cell just north of Svalbard using ensemble member 35: surface temperature (purple), thermodynamic ice volume tendency (navy blue) and snowmelt (brown). The X of each color corresponds to the defined melt onset date.

[Figure]

Figure R.3.: Average melt onset dates over 1979–2014 for each CESM LE definition using ensemble member 35: (a) snowmelt definition (b) thermodynamic ice volume tendency definition (c) surface temperature definition and (d) PMW satellite observations. The black line denotes the mean March ice edge (15% ice concentration) from 1979–2014 using (a–c) the CESM LE and (d) NSIDC Bootstrap sea ice concentrations (Comiso, 2017). Melt onset dates south of the mean ice edge are less reliable than those north of the edge.